# LEARNING WITH USER-LEVEL LOCAL DIFFERENTIAL PRIVACY

## ABSTRACT

User-level privacy is important in distributed systems. Previous research primarily focuses on the central model, while the local models have received much less attention. Under the central model, user-level DP is strictly stronger than the item-level one. However, under the local model, the relationship between user-level and item-level LDP becomes more complex, thus the analysis is crucially different. In this paper, we first analyze the mean estimation problem and then apply it to stochastic optimization, classification, and regression. In particular, we propose adaptive strategies to achieve optimal performance at all privacy levels. Moreover, we also obtain information-theoretic lower bounds, which show that the proposed methods are minimax optimal up to logarithmic factors. Unlike the central DP model, where user-level DP always leads to slower convergence, our result shows that under the local model, the convergence rates are nearly the same between user-level and item-level cases for distributions with bounded support. For heavy-tailed distributions, the user-level rate is even faster than the item-level one.

## 1 INTRODUCTION

Differential privacy (DP) (Dwork et al., 2006) is one of the mainstream schemes for privacy protection. The traditional DP framework is item-level, which focuses on the privacy of each sample (Dwork et al., 2014). However, in many real-world scenarios such as federated learning (Kairouz et al., 2021; Geyer et al., 2017; McMahan et al., 2018; Wang et al., 2019; Wei et al., 2020; 2021; Huang et al., 2023), each user provides multiple samples, which need to be treated as a whole for privacy protection. Therefore, in recent years, user-level differential privacy has emerged and has received widespread attention from researchers (Liu et al., 2020; Levy et al., 2021; Ghazi et al., 2021; 2023; Liu & Asi, 2024; Acharya et al., 2023; Zhao et al., 2024).

Existing research on user-level DP mainly focuses on central models, while local models have received relatively less attention. Several existing works have proposed algorithms for some learning tasks under user-level $\epsilon$-LDP with $\epsilon < 1$, based on samples from bounded domains. For example, (Bassily & Sun, 2023) focus on the stochastic optimization problem, and (Ma et al., 2024) provides a new method for sparse linear regression. Despite such progresses, there are still some remaining challenges. The most important one is that in industrial applications, from an accuracy-first perspective, a practical LDP protocol in a real-world LDP data collection system usually requires $\epsilon \geq 1$ (Talwar et al., 2024; Apple & Google, 2021). To achieve a good privacy-utility tradeoff, a fixed LDP protocol is not uniformly suitable to all $\epsilon$, thus a unified strategy adaptive to all privacy levels is crucially needed. Apart from the adaptivity to all privacy budgets, there are some other remaining problems. Real-world applications often involve samples generated from heavy-tailed distributions (Ibragimov et al., 2015), thus we need to handle the tails properly. Moreover, some important problems including nonparametric classification and regression have not been analyzed before.

In this paper, we conduct a systematic study on user-level $\epsilon$-LDP for a wide range of statistical tasks. We analyze the mean estimation problem first, including one-dimensional and multi-dimensional cases. We then apply the mean estimation methods to other tasks, including stochastic optimization, classification, and regression. For each task, we provide algorithms and analyze the theoretical convergence rates. Moreover, we derive the information-theoretic lower bounds based on classical minimax theory (Tsybakov, 2009), which shows that the newly proposed methods are minimax rate optimal up to a logarithm factor. The results are shown in Table 1, in which the non-private term

Table 1: Comparison of performance under user-level and item-level LDP.

| Tasks | user-level $n$ users, $m$ samples per user | item-level $nm$ samples |
|---|---|---|
| Mean, bounded | $\tilde{O}\left(\frac{d}{nm(\epsilon^2 \wedge \epsilon)}\right)$ $(n(\epsilon^2 \wedge 1) \gtrsim d\ln m$, Theorem 5) | $O\left(\frac{d}{nm(\epsilon^2 \wedge \epsilon)}\right)$ (Asi et al., 2022) |
| Mean, heavy-tail | $\tilde{O}\left(\frac{d\ln m}{mn(\epsilon^2 \wedge \epsilon)} \vee \left(\frac{d}{m^2 n(\epsilon^2 \wedge \epsilon)}\right)^{1-\frac{1}{p}}\right)$ $(n(\epsilon^2 \wedge 1) \gtrsim d\ln m$, Remark 2) | $O\left(\left(\frac{d}{nm(\epsilon^2 \wedge \epsilon)}\right)^{1-\frac{1}{p}}\right)$ (Duchi et al., 2018) [1] |
| Stochastic optimization | $\tilde{O}\left(\sqrt{\frac{d}{nm(\epsilon^2 \wedge \epsilon)}}\right)$ $(n(\epsilon^2 \wedge 1) \gtrsim d\ln n\ln m$, Theorem 8 | $\tilde{O}\left(\sqrt{\frac{d}{nm(\epsilon^2 \wedge \epsilon)}}\right)$ (Duchi et al., 2013) [2] |
| Classification | $\tilde{O}\left((mn(\epsilon^2 \wedge \epsilon))^{-\frac{\beta(1+\gamma)}{2(d+\beta)}}\right)$ $(n(\epsilon^2 \wedge 1) \gtrsim d\ln(mn)$, Theorem 9) | $O\left((mn(\epsilon^2 \wedge \epsilon))^{-\frac{\beta(1+\gamma)}{2(d+\beta)}}\right)$ (Berrett & Butucea, 2019) |
| Regression | $\tilde{O}\left((mn(\epsilon^2 \wedge \epsilon))^{-\frac{\beta}{d+\beta}}\right)$ $(n(\epsilon^2 \wedge 1) \gtrsim d\ln(mn)$, Theorem 11) | $O\left((mn(\epsilon^2 \wedge \epsilon))^{-\frac{\beta}{d+\beta}}\right)$ (Berrett et al., 2021) |

is omitted for simplicity. Under central DP, user-level DP is strictly stronger than item-level one, and thus always leads to a slower convergence rate (Levy et al., 2021). On the contrary, under the local model, the same convergence rates are derived between user-level and item-level cases for distributions with bounded support. If the distribution is heavy-tailed, then perhaps surprisingly, the user-level rate is even faster, such as those shown in the second row in Table 1.

The aforementioned challenges are addressed as follows. Firstly, we design algorithms that are adaptive to all privacy levels, which is especially important for multi-dimensional mean estimation. For $\epsilon < 1$, we conduct *user splitting* which divides users into groups and each group is used to estimate only one component. For very large $\epsilon$, we conduct *budget splitting*, which assign each component with privacy budget $\epsilon/d$. In the medium privacy regime, we design the splitting strategy carefully to achieve a smooth transition between these two extremes. Such design enables us to handle all privacy levels. Moreover, for heavy-tailed distributions, we clip each sample properly and achieve a good bias-variance tradeoff based on our theoretical analysis.

The main contributions of this paper are summarized as follows.

- For the mean estimation problem, we use a two-stage approach for $d = 1$. With higher dimensionality, for $\ell_\infty$ support, our method divides users into groups, and the strategy of such grouping is tailored to the privacy level $\epsilon$. We then use Kashin's representation to obtain a tight result for $\ell_2$ support.

- We apply the mean estimation to the stochastic optimization problem and derive a rate of $\tilde{O}(d/(nm(\epsilon^2 \wedge \epsilon)))$, matches the item-level bound in (Duchi et al., 2013) under the same total sample size.

- For nonparametric classification and regression, we divide the support into grids and apply the Hadamard transform, which is shown to be optimal under user-level LDP.

In general, the results show that the user-level LDP requirement is similar or sometimes even weaker than the item-level one, which is crucially different from the central model.

## 2 RELATED WORK

**Item-level DP.** We start with mean estimation, which is a basic but important statistical task since it serves as building blocks of stochastic optimization and deep learning (Abadi et al., 2016), which requires estimating the mean of gradients. (Asi & Duchi, 2020; Bun & Steinke, 2019; Huang et al., 2021; Liu et al., 2021; Hopkins et al., 2022; Vargaftik et al., 2021) studied mean estimation under

---

[1] For heavy-tailed distribution, (Duchi et al., 2018) analyzed the one dimensional case. We generalize it to $d$ dimensions.

[2] (Duchi et al., 2013) analyzed the case with $\epsilon \leq 1/4$. We generalize it to larger $\epsilon$.

central DP. For the local model, (Duchi et al., 2018) introduces an order optimal mean estimation method, which is then improved in (Li et al., 2023; Feldman & Talwar, 2021). Moreover, (Chen et al., 2020) achieved optimal communication cost. (Bhowmick et al., 2018) proposed PrivUnit, which is then shown to be optimal in constants (Asi et al., 2022). (Asi et al., 2024a) proposed ProjUnit, which reduces the communication complexity of PrivUnit. Mean estimation can be used in other problems. For example, in stochastic optimization, various methods have been proposed under central DP requirements (Chaudhuri et al., 2011; Bassily et al., 2014; 2019; Feldman et al., 2020; Asi et al., 2021; Kamath et al., 2022). Under local DP, (Duchi et al., 2013) proposed a stochastic gradient method, which calculates the noisy gradient from each sample and then update the model. For nonparametric statistics, (Duchi et al., 2013; 2018) shows that the nonparametric density estimation under LDP has a convergence rate of $O((n\epsilon^2)^{-\beta/(d+\beta)})$ for small $\epsilon$, which is inevitably slower than the non-private rate $O(n^{-\beta/(d+2\beta)})$ (Tsybakov, 2009). (Berrett & Butucea, 2019) and (Berrett et al., 2021) extend the analysis to nonparametric classification and regression problems, respectively. Moreover, several works extend the analysis of DP to sparse settings (Zhu et al., 2023; Zhou et al., 2022)

**User-level DP.** Under central model, (Geyer et al., 2017) proposes a simple clipping method. (Levy et al., 2021) designs a two-stage approach for one-dimensional mean estimation, and then extends to higher dimension using the Hadamard transform. (Cummings et al., 2022) studies mean estimation under data heterogeneity. This method is then used in stochastic optimization problems (Bassily & Sun, 2023; Liu & Asi, 2024). Additionally, some works are focusing on black-box conversion from item-level DP to user-level, such as (Ghazi et al., 2021; Bun et al., 2023; Ghazi et al., 2023). (Li et al., 2024; Charles et al., 2024) apply user-level DP in deep learning. Under the local model, (Acharya et al., 2023) studies the discrete distribution estimation problem. (Bassily & Sun, 2023) studies the optimization problem with $\epsilon < 1$. (Ma et al., 2024) analyzes the linear regression problem.

Compared with existing works, to the best of our knowledge, our work is the first attempt to analyze mean estimation and stochastic optimization problems under user-level LDP for general $\epsilon$. Unlike the central model, in which a single algorithm structure is enough, we have to design adaptive privacy mechanisms that are tailored to every possible $\epsilon$ under the local model. Moreover, we also provide the first analysis on nonparametric classification and regression problems under user-level $\epsilon$-LDP.

## 3 PRELIMINARIES

Suppose there are $n$ users, and each user has $m$ identical and independently distributed (i.i.d) samples, denoted as $\mathbf{X}_{ij} \in \mathcal{X}$, $i = 1, \ldots, n$, $j = 1, \ldots, m$. Let $\mathbf{X}_i = \{\mathbf{X}_{i1}, \ldots, \mathbf{X}_{im}\}$ be the set of all samples stored in user $i$. Due to privacy concerns, users are unwilling to upload $\mathbf{X}_i$ directly. Instead, there is a privacy mechanism that transforms $\mathbf{X}_1, \ldots, \mathbf{X}_n$ into $n$ random variables $\mathbf{Z}_1, \ldots, \mathbf{Z}_n \in \mathcal{Z}$ with $\mathbf{Z}_i = M_i(\mathbf{X}_i, \mathbf{Z}_1, \ldots, \mathbf{Z}_{i-1})$, in which $M_i : \mathcal{X} \times \mathcal{Z}^{i-1} \to \mathcal{Z}$ is a function with random output. The user-level LDP is defined as follows.

**Definition 1.** *Given a privacy parameter $\epsilon \geq 0$, the privacy mechanism $M_i$ is user-level $\epsilon$-LDP if for all $i$, all values of $\mathbf{z}_1, \ldots, \mathbf{z}_{i-1}$, all $\mathbf{x}, \mathbf{x}' \in \mathcal{X}^m$ and all $S \subseteq \mathcal{Z}$,*

$$P(\mathbf{Z}_i \in S | \mathbf{X}_i = \mathbf{x}, \mathbf{Z}_{1:i-1} = \mathbf{z}_{1:i-1}) \leq e^{\epsilon} P(\mathbf{Z}_i \in S | \mathbf{X}_i = \mathbf{x}', \mathbf{Z}_{1:i-1} = \mathbf{z}_{1:i-1}), \tag{1}$$

*in which $\mathbf{Z}_i = M_i(\mathbf{X}_i, \mathbf{Z}_1, \ldots, \mathbf{Z}_{i-1})$, $\mathbf{Z}_{1:i-1} = (\mathbf{Z}_1, \ldots, \mathbf{Z}_{i-1})$, $\mathbf{z}_{1:i-1} = (\mathbf{z}_1, \ldots, \mathbf{z}_{i-1})$.*

Definition 1 requires that the distributions of $\mathbf{Z}_i$ should not change much even if the whole local dataset $\mathbf{X}_i = \{\mathbf{X}_{i1}, \ldots, \mathbf{X}_{im}\}$ is altered. From (1), even if the adversary can observe $\mathbf{Z}_i$, it can not infer the value of $\mathbf{X}_i$ exactly. Smaller $\epsilon$ indicates stronger privacy protection since it is harder to distinguish $\mathbf{X}_i$. The difference between item-level and user-level LDP is illustrated in Figure 1. In the item-level case, each sample is transformed into a privatized one, while in the user-level case, all samples of a user

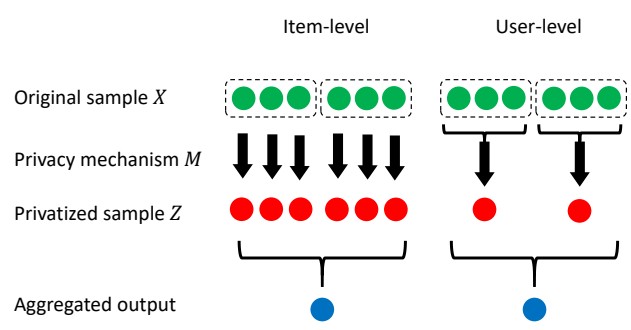

Figure 1: Comparison of item-level versus user-level LDP. Dashed rectangles represent users.

are combined to generate a privatized sample. For both item-level and user-level cases, at the final step, all privatized samples are aggregated to generate the output. One natural question is how the difficulty of achieving user-level LDP compares with the item-level counterparts. Regarding this question, we have the following statements.

**Proposition 1.** *Based on Definition 1, for any statistical problems, there are two basic facts:*

*(1) If item-level $(\epsilon/m)$-LDP can be achieved with $nm$ samples, then user-level $\epsilon$-LDP can be achieved using $n$ users with $m$ samples per user;*

*(2) If item-level $\epsilon$-LDP can be achieved with $n$ samples, then user-level $\epsilon$-LDP can be achieved using $n$ users with $m$ samples per user.*

In the above statements, (1) holds due to the group privacy property. For (2), if a task can be solved using $n$ samples under item-level $\epsilon$-LDP, then just randomly picking a sample from each user satisfies user-level $\epsilon$-LDP. These results also suggest two baseline methods that transform item-level methods to user-level. However, these simple conversions are far from optimal. For the first one, $(\epsilon/m)$-LDP is too strong. For the second one, many samples are wasted.

One may wonder if user-level LDP is a stronger requirement than the item-level one. In other words, if item-level $\epsilon$-LDP can be achieved with $nm$ samples, then can we achieve user-level $\epsilon$-LDP using $n$ users with $m$ samples per user? Under the central model, the answer is affirmative: user-level DP is stronger because the definition of user-level $\epsilon$-DP ensures item-level $\epsilon$-DP (Levy et al., 2021). Nevertheless, under the local model, things become more complex. On the one hand, user-level LDP imposes stronger privacy requirements, since the distribution of the output variables can only change to a limited extent even when the local dataset is replaced as a whole. On the other hand, user-level LDP enables samples within the same user to share information with each other, thus the difficulty is somewhat reduced in this aspect. From Table 1, for many problems, with the same total sample sizes, user-level and item-level LDP yield nearly the same error bounds. If the distribution has tails, then the user-level LDP is even easier to achieve, which is perhaps surprising.

Before discussing each task in detail, we clarify some notations that will be used in subsequent sections. Denote $a \wedge b = \min(a, b)$, $a \vee b = \max(a, b)$, and $a \lesssim b$ if there exists a constant $C$ that may depend on the constants made in problem assumptions, such that $a \le Cb$. Conversely, $a \gtrsim b$ means $a \ge Cb$. $a \sim b$ means that $a \lesssim b$ and $a \gtrsim b$ both hold.

## 4 MEAN ESTIMATION

For one-dimensional problem, we introduce a two-stage method. Despite that similar idea has also been used in central user-level DP (Levy et al., 2021), details and theoretical analysis are different. We then extend the analysis to high-dimensional problems. To achieve optimal convergence rate for all privacy levels, our strategies are designed separately for each $\epsilon$.

### 4.1 ONE DIMENSIONAL CASE

We start with the case such that the distribution has bounded support $\mathcal{X} = [-D, D]$ for some $D$, and introduce a two-stage method. The first stage uses half of the users to identify an interval $[L, R]$, which is much smaller than $[-D, D]$ but contains $\mu := \mathbb{E}[X]$ with high probability. The purpose of this stage is to significantly reduce the

---

**Algorithm 1** MeanEst1d: One dimensional mean estimation under user-level $\epsilon$-LDP

**Input:** Dataset containing $n$ users with $m$ samples per user, i.e. $X_{ij}$, $i = 1, \dots, n$, $j = 1, \dots, m$
**Output:** Estimated mean $\hat{\mu}$
**Parameter:** $h$, $\Delta$, $D$, $\epsilon$

1: Calculate $Y_i = (1/m) \sum_{j=1}^{m} X_{ij}$ for $i = 1, \dots, n/2$;
2: Divide $[-D, D]$ into $B$ bins of length $h$;
3: $Z_{ik} = \mathbf{1}(Y_i \in B_k) + W_{ik}$ for $i = 1, \dots, n/2$, $k = 1, \dots, B$, in which $W_{ik} \sim \text{Lap}(2/\epsilon)$;
4: Calculate $s_k = \sum_{i=1}^{n/2} Z_{ik}$ for $k = 1, \dots, B$;
5: Let $\hat{k}^* = \arg\max_k s_k$;
   $L = -D + (\hat{k}^* - 2)h$;
   $R = -D + (\hat{k}^* + 1)h$;
6: $Z_i = (Y_i \vee (L - \Delta)) \wedge (R + \Delta) + W_i$ for $i = n/2 + 1, \dots, n$, in which $W_i \sim \text{Lap}((3h + 2\Delta)/\epsilon)$;
7: Calculate $\hat{\mu} = (2/n) \sum_{i=n/2+1}^{n} Z_i$;
8: **Return** $\hat{\mu}$

---

strength of Laplacian noise needed to
protect privacy, and thus reduce the neg-
ative effect on the estimation accuracy
caused by privacy mechanisms. At the second stage, the algorithm then truncates the values into $[L, R]$, and adds a Laplacian noise to ensure $\epsilon$-LDP at user-level. Finally, $\mu$ can be estimated with a simple average over the other half of users. The details are provided in Algorithm 1.

The privacy guarantee and the estimation error of Algorithm 1 are both analyzed in Theorem 1. In Algorithm 1, $\mathrm{Lap}(\lambda)$ means Laplacian distribution with parameter $\lambda$, whose probability density function (pdf) is $f(u) = e^{-|u|/\lambda}/(2\lambda)$.

**Theorem 1.** *Algorithm 1 is user-level $\epsilon$-LDP. If $n(\epsilon^2 \wedge 1) \geq c_1 \ln m$ for a constant $c_1$, then with $h = 4D/\sqrt{m}$ and $\Delta = D\sqrt{\ln n/m}$, the mean squared error of one dimensional mean estimation under user-level $\epsilon$-LDP satisfies*

$$\mathbb{E}[(\hat{\mu} - \mu)^2] \lesssim \frac{D^2}{nm}\left(1 + \frac{\ln n}{\epsilon^2}\right). \tag{2}$$

The proof of Theorem 1 is shown in Appendix A. In Appendix A.2, we show that $[L, R]$ contains $\mu$ with high probability. To begin with, $\hat{k}^* \in \{k^* - 1, k^*, k^* + 1\}$ holds with high probability, in which $k^*$ is the index of the bin containing $\mu$, i.e. $\mu \in B_{k^*}$. Let $L$ be the left bound of the $(\hat{k}^* - 1)$-th bin, and $R$ be the right bound of the $(\hat{k}^* + 1)$-th bin, then with high probability, $\mu \in [L, R]$. In Appendix A.3, we then bound the bias and variance separately. As shown in Proposition 1, there are two baseline methods to achieve user-level LDP from item-level LDP. The first one is to achieve item-level $(\epsilon/m)$-LDP for all samples. This yields a bound $O(D^2 m/(n\epsilon^2) + D^2/(nm))$. The second one is to achieve item-level $\epsilon$-LDP for $n$ samples randomly selected from $n$ users, which also only yields $O(D^2/(n(\epsilon^2 \wedge 1)))$, significantly worse than the right hand side of (2).

In Theorem 1, the requirement $n(\epsilon^2 \wedge 1) \geq c_1 \ln m$ is necessary since if $n$ is fixed, then the mean squared error will never converge to zero with increasing $m$. From an information-theoretic perspective, a fixed number of privatized variables can only transmit limited information (Cuff & Yu, 2016; Wang et al., 2016). Therefore, it is necessary to let $n$ grow with $m$, which is also discussed in (Levy et al., 2021) for user-level central DP. Theorem 2 shows the information-theoretic minimax lower bound.

**Theorem 2.** *Denote $\mathcal{P}_{\mathcal{X}}$ as the set of all distributions supported on $\mathcal{X} = [-D, D]$, $\mathcal{M}_{\epsilon}$ as all mechanisms satisfying $\epsilon$-LDP, then*

$$\inf_{\hat{\mu}} \inf_{M \in \mathcal{M}_{\epsilon}} \sup_{p \in \mathcal{P}_{\mathcal{X}}} \mathbb{E}\left[(\hat{\mu} - \mu)^2\right] \gtrsim \frac{D^2}{nm(\epsilon^2 \wedge 1)}. \tag{3}$$

*Moreover, with fixed $n$, the mean squared error will not converge to zero as $m$ increases. To be more precise, $\mathbb{E}[(\hat{\mu} - \mu)^2] \geq (1/4)D^2 e^{-n\epsilon(e^{\epsilon}-1)}$.*

The proof of Theorem 2 is shown in Appendix A.4. The comparison between (2) and (3) shows that the upper and lower bounds match up to a logarithm factor, thus the two-stage method is nearly minimax optimal. Finally, we extend the method to the case with unbounded support. In this case, we replace step 1 in Algorithm 1 with $Y_i = -D \vee (\bar{X}_i \wedge D)$, in which $\bar{X}_i = (1/m)\sum_{j=1}^{m} X_{ij}$ is the $i$-th user-wise mean. Such clipping operation controls the sensitivity. Other steps are the same as Algorithm 1. The convergence rate is shown in Theorem 3.

**Theorem 3.** *Assume that $\mathbb{E}[|X|^p] \leq M_p < \infty$ for some finite constant $M_p$, with $p \geq 2$. If $n(\epsilon^2 \wedge 1) \geq c_1 \ln m$, then with Algorithm 1, except that step 1 is replaced by $Y_i = -D \vee (\bar{X}_i \wedge D)$, the mean squared error of $\hat{\mu}$ can be bounded by*

$$\mathbb{E}[(\hat{\mu} - \mu)^2] \lesssim M_p^{2/p}\left[\frac{\ln m}{mn\epsilon^2} \vee (m^2 n\epsilon^2)^{-\left(1 - \frac{1}{p}\right)} + \frac{1}{mn}\right]. \tag{4}$$

The selection of $D$ and the proof of Theorem 3 are shown in Appendix A.5. Here we provide an intuitive understanding of the phase transition in (4). As long as $p \geq 2$, from central limit theorem, with large $m$, similar to the case with bounded support, $Y_i$ is nearly normally distributed, and the tail is like a Gaussian distribution. Therefore, the convergence rate of the mean squared error is still

$O(\ln m/(mn\epsilon^2))$, the same as the case with bounded support. However, if $m$ is small, the Gaussian approximation no longer holds. In this case, the tail of the distribution of $Y_i$ is polynomial. As a result, there is a phase transition in (4). Mean estimation for heavy-tailed distributions is an example that user-level LDP is easier to achieve than the item-level one. With $nm$ samples, mean squared error under item-level $\epsilon$-LDP is $O((nm\epsilon^2)^{1-1/p})$ (Duchi et al., 2018), significantly worse than (4).

### 4.2 MULTI-DIMENSIONAL CASE

This section discusses the mean estimation problem with $d \geq 1$. Depending on the shape of the support set, the problem can be crucially different. Here we discuss two cases, i.e. $\ell_2$ support $\mathcal{X}_2 = \{\mathbf{u}|\, \|\mathbf{u}\|_2 \leq D\}$, and $\ell_\infty$ support $\mathcal{X}_\infty = \{\mathbf{u}|\, \|\mathbf{u}\|_\infty \leq D\}$. For small $\epsilon$, the mean squared error under item-level $\epsilon$-LDP is $O(d/(n(\epsilon^2 \wedge \epsilon)))$ for $\ell_2$ support, and $O(d^2/(n(\epsilon^2 \wedge \epsilon)))$ for $\ell_\infty$ support (Duchi et al., 2018; Asi et al., 2022; Feldman & Talwar, 2021; Asi et al., 2024a). Similar to the one-dimensional case, direct transformation to user-level according to Proposition 1 yields a suboptimal bound.

$\ell_\infty$ **Support.** To begin with, we focus on this relatively simpler case. The method depends on the value of $\epsilon$. Details are stated in Algorithm 2.

*1) High privacy ($\epsilon < 1$).* Users are assigned randomly into $d$ groups, and the $k$-th group is used to estimate $\mu_k$ (the $k$-th component of $\mu := \mathbb{E}[\mathbf{X}]$) for $k = 1, \ldots, K$. Since the size of each group is $n/d$, from (2), we have

$$\mathbb{E}[(\hat{\mu}_k - \mu_k)^2] \quad \lesssim \quad \frac{D^2}{(n/d)m}\left(1 + \frac{\ln(n/d)}{\epsilon^2}\right) \lesssim \frac{D^2 d \ln n}{nm(\epsilon^2 \wedge 1)}. \tag{5}$$

---

**Algorithm 2** MeanEst: Multi-dimensional mean estimation under user-level $\epsilon$-LDP with $\ell_\infty$ support

**Input:** Dataset containing $n$ users with $m$ samples per user, i.e. $\mathbf{X}_{ij}$, $i = 1, \ldots, n$, $j = 1, \ldots, m$
**Output:** Estimated mean $\hat{\mu}$
**Parameter:** $h$, $\Delta$, $D$, $\epsilon$
  1: **if** $\epsilon < 1$ **then**
  2:     Divide users randomly into $d$ groups $S_1, \ldots, S_d$;
  3:     **for** $k = 1, \ldots, d$ **do**
  4:         Estimate $\hat{\mu}_k$ with $S_k$ using Algorithm 1 for $k = 1, \ldots, d$ under $\epsilon$-LDP;
  5:     **end for**
  6: **else if** $1 \leq \epsilon < d \ln n$ **then**
  7:     Divide users into $\lceil d/\epsilon \rceil$ groups $S_1, \ldots, S_{\lceil d/\epsilon \rceil}$;
  8:     **for** $k = 1, \ldots, \lceil d/\epsilon \rceil$ **do**
  9:         **for** $l = (k-1)\epsilon + 1, \ldots, k\epsilon \wedge d$ **do**
 10:             Estimate $\hat{\mu}_l$ with $S_k$ using Algorithm 1 under 1-LDP;
 11:         **end for**
 12:     **end for**
 13: **else**
 14:     **for** $k = 1, \ldots, d$ **do**
 15:         Estimate $\hat{\mu}_k$ with all users using Algorithm 1 under $(\epsilon/d)$-LDP
 16:     **end for**
 17: **end if**
 18: **return** $\hat{\mu} = (\hat{\mu}_1, \ldots, \hat{\mu}_d)$

---

*2) Medium privacy ($1 \leq \epsilon < d \ln n$).* In this case, the privacy requirement is weaker than the case with $\epsilon < 1$. Therefore, a group of users can be used to estimate more components, with $\epsilon$-LDP still satisfied. Without loss of generality, suppose that $\epsilon$ is an integer (otherwise one can just strengthen the requirement to $\lfloor \epsilon \rfloor$-LDP). In this case, users are randomly allocated to $\lceil d/\epsilon \rceil$ groups. Each group is used to estimate $\epsilon$ components, and each component is estimated under user-level 1-LDP. From basic composition theorem (Dwork et al., 2010), estimating $\epsilon$ components of $\mu$ satisfies user-level $\epsilon$-LDP. Denote $n_0$ as the number of users in each group, then

$$\mathbb{E}[(\hat{\mu}_k - \mu_k)^2] \lesssim \frac{D^2}{n_0 m}(1 + \ln n_0) \sim \frac{D^2 \ln(n\epsilon/d)}{(n\epsilon/d)m} \lesssim \frac{D^2 d}{nm\epsilon} \ln n. \tag{6}$$

In the first step, we replace $n$ and $\epsilon$ in (2) with $n_0$ and 1 respectively, since now we are using a group with $n_0$ users to achieve 1-LDP.

*3) Low privacy ($\epsilon \geq d \ln n$).* In this case, the privacy protection is much less important. We hope that the estimation error is as close to the non-private case as possible. Based on such intuition, we no longer divide users into groups. Instead, our method just estimates each component under user-level $(\epsilon/d)$-LDP, then the whole algorithm is $\epsilon$-LDP. In this case, the mean squared error of each component is bounded by

$$\mathbb{E}[(\hat{\mu}_k - \mu_k)^2] \lesssim \frac{D^2}{nm}\left(1 + \frac{d^2 \ln n}{\epsilon^2}\right) \lesssim \frac{D^2}{nm}. \tag{7}$$

Note that $\mathbb{E}[\|\hat{\mu} - \mu\|^2] \leq \sum_{k=1}^{d} \mathbb{E}[(\hat{\mu}_k - \mu_k)^2]$. A combination (5), (6) and (7) yields the following theorem.

**Theorem 4.** *Under user-level $\epsilon$-LDP, if $n(\epsilon^2 \wedge 1) \geq c_1 d \ln m$, in which $c_1$ is the constant in Theorem 1, then the mean squared error of multi-dimensional mean estimation in $\mathcal{X}_\infty$ with Algorithm 2 is bounded by*

$$\mathbb{E}\left[\|\hat{\mu} - \mu\|_2^2\right] \lesssim \frac{D^2 d}{nm}\left(1 + \frac{d \ln n}{\epsilon^2 \wedge \epsilon}\right). \tag{8}$$

We would like to remark that under central DP, the loss caused by privacy mechanisms and the non-private loss are two separate terms, and we only need to select the aggregator to minimize the latter one, which does not depend on $\epsilon$. However, under the local model, privatization takes place before aggregation. Depending on $\epsilon$, the optimal randomization can be crucially different. Therefore, it is necessary to discuss each $\epsilon$ separately. In Theorem 4, we give a complete picture of the estimation error caused by different privacy levels. In particular, with $\epsilon \to \infty$, (8) converges to $D^2 d/(nm)$, which is just the non-private rate.

$\ell_2$ **Support.** Consider that $\ell_2$ support is smaller than the $\ell_\infty$ support, we expect that the bound of mean squared error can be improved over (8). Directly applying Algorithm 2 does not make any improvement. Therefore, a more efficient approach is needed to achieve a better bound. Towards this goal, we use Kashin's representation (Lyubarskii & Vershynin, 2010), which has also been used in other problems related to stochastic estimation (Feldman et al., 2021; Chen et al., 2023; Asi et al., 2024b). To begin with, we rephrase Kashin's representation as follows.

**Lemma 1.** *(Kashin's representation, rephrased from Theorem 2.2 in (Lyubarskii & Vershynin, 2010)) There exists a matrix $\mathbf{U} \in \mathbb{R}^{2d \times d}$ and a constant $K$, such that $\mathbf{U}^T \mathbf{U} = \mathbf{I}_d$, in which $I_d$ is the $d \times d$ identity matrix, and for all $\mathbf{x}$ with $\|\mathbf{x}\|_2 \leq 1$, $\|\mathbf{U}\mathbf{x}\|_\infty \leq K/\sqrt{d}$.*

Based on Lemma 1, our method constructs matrix $\mathbf{U} = (\mathbf{u}_1, \ldots, \mathbf{u}_{2d})^T \in \mathbb{R}^{2d \times d}$. Then we can transform all samples. Let $\mathbf{X}'_{ij} = \mathbf{U}\mathbf{X}_{ij}$ for $i = 1, \ldots, n, j = 1, \ldots, m$. Correspondingly, denote $\theta = \mathbf{U}\mu$ as the mean vector after transformation. Then $\mu$ can be estimated by estimating $\theta$ first. Since $\mathbf{X}_{ij} \in \mathcal{X}_2$, $\|\mathbf{X}_{ij}\|_2 \leq D$ holds. According to Lemma 1, $\left\|\mathbf{X}'_{ij}\right\|_\infty \leq KD/\sqrt{d}$. Therefore, we have transformed the $\ell_2$ support into $\ell_\infty$ support, thus $\theta$ can be estimated using Algorithm 2. The only difference is that now the supremum norm is reduced from $D$ to $KD/\sqrt{d}$. After getting $\hat{\theta}$, we then transform it back to $\ell_2$ support, i.e. $\hat{\mu} = \mathbf{U}^T \hat{\theta}$. Since $\hat{\theta}$ is user-level $\epsilon$-LDP, it is guaranteed that $\hat{\mu}$ is also user-level $\epsilon$-LDP. The following theorem bounds the mean squared error of $\hat{\mu}$.

**Theorem 5.** *Under user-level $\epsilon$-LDP, if $n(\epsilon^2 \wedge 1) \geq c_1 d \ln m$, then the mean squared error of multi-dimensional mean estimation in $\mathcal{X}_2$ is bounded by*

$$\mathbb{E}\left[\|\hat{\mu} - \mu\|_2^2\right] \lesssim \frac{D^2}{nm}\left(1 + \frac{d \ln n}{\epsilon^2 \wedge \epsilon}\right). \tag{9}$$

The proof of Theorem 5 is shown in Appendix B.1.

**Remark 1.** *If the support is $\ell_1$, then we can also let $\mathbf{U} = \mathbf{H}_d/\sqrt{d}$, in which $\mathbf{H}_d$ is the $d \times d$ Hadamard matrix (Hedayat & Wallis, 1978). This can be used in the discrete distribution estimation problem. With alphabet size $A$, each sample $\mathbf{X}_{ij}$ can be viewed as a $A$ dimensional vector, such that $\mathbf{X}_{ijk} = 1$ for some $k$ and $\mathbf{X}_{ijl} = 0$ for $k \neq l$. Then the $\ell_2$ estimation error is bounded by $O(A \ln n/(nm(\epsilon^2 \wedge \epsilon)))$, which matches (Acharya et al., 2023) up to logarithm factor.*

The corresponding minimax lower bounds are shown as follows.

**Theorem 6.** *Denote $\mathcal{P}_{\mathcal{X},p}$ as the set of all distributions supported on $\mathcal{X}_p = \{\mathbf{u} | \|\mathbf{u}\|_p \leq D\}$, $\mathcal{M}_\epsilon$ as all mechanisms satisfying user-level $\epsilon$-LDP. Then for $p \in [1, 2]$, with $n$ users and $m$ samples per user,*

$$\inf_{\hat{\mu}} \inf_{M \in \mathcal{M}_\epsilon} \sup_{p \in \mathcal{P}_{\mathcal{X},p}} \mathbb{E}\left[\|\hat{\mu} - \mu\|_2^2\right] \gtrsim \frac{D^2 d}{nm(\epsilon^2 \wedge \epsilon)}. \tag{10}$$

**Theorem 7.** *Denote $\mathcal{P}_{\mathcal{X},\infty}$ as the set of all distributions supported on $\mathcal{X}_\infty$, $\mathcal{M}_\epsilon$ as all mechanisms satisfying $\epsilon$-LDP. Then with $n$ users and $m$ samples per user,*

$$\inf_{\hat{\mu}} \inf_{M \in \mathcal{M}_\epsilon} \sup_{p \in \mathcal{P}_{\mathcal{X},\infty}} \mathbb{E}\left[\|\hat{\mu} - \mu\|_2^2\right] \gtrsim \frac{D^2 d^2}{nm(\epsilon^2 \wedge \epsilon)}. \tag{11}$$

The proof of Theorem 6 and 7 are shown in Appendix B.2 and B.3, respectively. The upper bounds (8) and (9) match the lower bounds (11) and (10). These results indicate that our methods for high dimensional mean estimation under user-level LDP are minimax optimal.

**Remark 2.** *Now we extend the analysis to unbounded support. If $\mathbb{E}[|X_k|^p] \leq M_p$ for all $k = 1, \ldots, d$, then with $n\epsilon^2 \geq c_1 d \ln m$ for some constant $c_1$,*

$$\mathbb{E}\left[\|\hat{\mu} - \mu\|_2^2\right] \lesssim M_p^{2/p} \left[\frac{d^2 \ln m}{mn(\epsilon^2 \wedge \epsilon)} \vee \left(\frac{d}{m^2 n(\epsilon^2 \wedge \epsilon)}\right)^{1-1/p} + \frac{d}{mn}\right]. \tag{12}$$

*Under a stronger condition $\mathbb{E}[\|\mathbf{X}\|_2^p] \leq M_p < \infty$, the mean squared error can be bounded by*

$$\mathbb{E}\left[\|\hat{\mu} - \mu\|_2^2\right] \lesssim M_p^{2/p} \left[\frac{d \ln m}{mn(\epsilon^2 \wedge \epsilon)} \vee \left(\frac{d}{m^2 n(\epsilon^2 \wedge \epsilon)}\right)^{1-1/p} + \frac{1}{mn}\right], \tag{13}$$

*which is smaller than the rate under coordinate-wise $p$-th order bounded moment by a factor $d$. The detailed arguments can be found in Appendix B.4.*

## 5 STOCHASTIC OPTIMIZATION

The goal is to solve the following stochastic optimization problem. Define the loss function as $L(\theta) := \mathbb{E}[l(\mathbf{X}, \theta)]$, in which $\mathbf{X}$ is a random variable following distribution $p$. Given $\mathbf{X}_{ij}$, $i = 1, \ldots, n$, $j = 1, \ldots, m$, our goal is to find the minimizer

$$\theta^* = \min_{\theta \in \Theta} L(\theta). \tag{14}$$

The estimator is designed as follows. Users are divided randomly into $t_0$ groups. We plan to update $\theta$ in $t_0$ steps. In the $t$-th step, we use one group of users to get an estimate of $\nabla L(\theta_t) = \mathbb{E}[\nabla l(\mathbf{X}, \theta_t)]$ using Algorithm 2, which includes the privacy mechanism. The result is denoted as $\mathbf{g}_t$, and the update rule of $\theta$ is

$$\theta_{t+1} = \theta_t - \eta \mathbf{g}_t, \tag{15}$$

in which $\eta$ is the learning rate. Since Algorithm 2 satisfies $\epsilon$-LDP at user-level, and each user is only used once, the whole algorithm with $t_0$ steps also satisfies $\epsilon$-LDP.

These steps are summarized in Algorithm 3. In step 5, the MeanEst function refers to the multi-dimensional mean estimation method shown in Algorithm 2. Samples are privatized in this step. Therefore, Algorithm 3 satisfies user-level $\epsilon$-LDP.

Now we provide a theoretical analysis, which is based on the following assumptions.

---

**Algorithm 3** Stochastic optimization under user-level $\epsilon$-LDP

**Input:** Dataset containing $n$ users with $m$ samples per user, i.e. $\mathbf{X}_{ij}$, $i = 1, \ldots, n$, $j = 1, \ldots, m$

**Output:** Estimated $\hat{\theta}$

1: Initialize $\theta_0$;
2: Divide users into $t_0$ groups $S_0, \ldots, S_{T-1}$;
3: **for** $t = 0, 1, \ldots, t_0 - 1$ **do**
4:     Calculate $\nabla l(\mathbf{X}_{ij}, \theta_t)$ for $i \in S_t$, $j = 1, \ldots, m$;
5:     $\mathbf{g}_t = MeanEst(\{\nabla l(\mathbf{X}_{ij}, \theta_t) | i \in S_t, j \in [m]\})$;
6:     $\theta_{t+1} = \theta_t - \eta \mathbf{g}_t$;
7: **end for**
8: **Return** $\hat{\theta} = \theta_{t_0}$

---

**Assumption 1.** *(a) $l(\mathbf{X}, \theta)$ is $G$-smooth, i.e. $\nabla l(\mathbf{X}, \theta)$ is $G$-Lipschitz, in which $\nabla$ denotes the gradient with respect to $\theta$;*

*(b) For any $\theta$, the gradient of $l$ has bounded $\ell_2$ norm with probability 1, i.e. $\|\nabla l(\mathbf{X}, \theta)\|_2 \leq D$;*

*(c) $L$ is $\gamma$-strong convex.*

The theoretical bound is shown in the following theorem.

**Theorem 8.** *With $\eta \leq 1/G$, the $\ell_2$ error at $t$-th step can be bounded by*

$$\mathbb{E}\left[\|\theta_t - \theta^*\|_2\right] \leq \left(1 - \frac{1}{2}\eta\gamma\right)^t \|\theta_0 - \theta^*\|_2 + \frac{2D}{\gamma}\sqrt{\frac{Ct_0}{nm}\left(1 + \frac{d\ln n}{\epsilon^2 \wedge \epsilon}\right)}. \tag{16}$$

*From (16), there exists two constants $c_T$ and $C_T$, if $c_T \ln n \leq t_0 \leq C_T \ln n$, and $n(\epsilon^2 \wedge 1) \gtrsim d\ln n \ln m$, then the final estimate $\hat{\theta} = \theta_{t_0}$ satisfies*

$$\mathbb{E}\left[\left\|\hat{\theta} - \theta^*\right\|_2\right] \lesssim D\sqrt{\frac{\ln n}{nm}\left(1 + \frac{d\ln n}{\epsilon^2 \wedge \epsilon}\right)}. \tag{17}$$

The proof of Theorem 8 is provided in Appendix C. In (Duchi et al., 2013), it is shown that the bound for item-level case is $\tilde{O}(\sqrt{d/(n\epsilon^2)})$ for $\epsilon \leq 1/4$ with $n$ samples. Therefore, with the same total number of samples, our bound matches the result in (Duchi et al., 2013).

# 6 NONPARAMETRIC CLASSIFICATION AND REGRESSION

From now on, we focus on nonparametric learning problems under user-level local DP. In previous sections, the dataset contains $n$ users with $m$ samples per user, i.e. $\mathbf{X}_{ij}$, $i = 1, \ldots, n$, $j = 1, \ldots, m$. For nonparametric learning problems, apart from $\mathbf{X}_{ij}$, we also have the label $Y_{ij}$. Following (Berrett & Butucea, 2019; Berrett et al., 2021), which focuses on item-level classification and regression problems, suppose that $\mathbf{X}$ is supported in $[0, 1]^d$, which is made for simplicity. It can be generalized to arbitrary bounded support. Denote $(\mathbf{X}, Y)$ as a test sample i.i.d to training samples, and the output of the classifier is $\hat{Y}$.

## 6.1 CLASSIFICATION

The risk is defined as $R = P(\hat{Y} \neq Y)$. Define $\eta(\mathbf{x}) = \mathbb{E}[Y|\mathbf{X} = \mathbf{x}]$. Given the test sample at $\mathbf{x}$, the optimal classifier is $\hat{Y} = \text{sign}(\eta(\mathbf{x}))$. The corresponding optimal risk, called Bayes risk, is

$$R^* = P(\text{sign}(\eta(\mathbf{X})) \neq Y) = \frac{1}{2}\mathbb{E}[1 - |\eta(\mathbf{X})|]. \tag{18}$$

$\eta$ is unknown in practice. We have to learn $\eta$ from the training data. Therefore, in reality, there is inevitably a gap between the risk of a practical classifier and the Bayes risk. Such gap is called excess risk $R - R^*$. To improve the efficiency, we propose a method based on a transformation with Hadamard matrix (Hedayat & Wallis, 1978). We make some assumptions before stating our algorithm.

**Assumption 2.** *There exists constants $C_a$, $C_b$, $f_L$, such that*

*(a) For all $t > 0$, $P(|\eta(\mathbf{X})| < t) \leq C_a t^\gamma$;*

*(b) For all $\mathbf{x}, \mathbf{x}' \in \mathcal{X} = [0, 1]^d$, $|\eta(\mathbf{x}) - \eta(\mathbf{x}')| \leq C_b \|\mathbf{x} - \mathbf{x}'\|_2^\beta$;*

*(c) $f(\mathbf{x}) \geq f_L$ for all $\mathbf{x} \in \mathcal{X}$.*

(a) is commonly used in many existing literatures and is typically referred to as 'Tsybakov noise condition' (Audibert & Tsybakov, 2007; Chaudhuri & Dasgupta, 2014; Döring et al., 2017). (b) is the Hölder smoothness condition, which is commonly used in nonparametric statistics (Tsybakov, 2009). (c) is usually referred to as 'strong density assumption', which is also commonly made (Döring et al., 2017; Gadat et al., 2016). Our basic assumptions (a)-(c) are the same as (Berrett & Butucea, 2019), except that we are now considering user-level LDP, while (Berrett & Butucea, 2019) is about item-level LDP.

**Theorem 9.** *Under Assumption 2, if $n(\epsilon^2 \wedge 1) \geq c_2(\ln m + \ln n)$ for some constant $c_2$, then there exists a classifier (the algorithm is shown in Appendix D.1), such that*

$$R - R^* \lesssim (mn(\epsilon^2 \wedge \epsilon))^{-\frac{\beta(1+\gamma)}{2(d+\beta)}} \ln^{1+\gamma} n + \left(\frac{nm}{\ln n}\right)^{-\frac{\beta(1+\gamma)}{2\beta+d}}. \tag{19}$$

The proof of Theorem 9 is shown in Appendix D.2. With large $\epsilon$, (19) reduces to $(mn/\ln n)^{-2\beta/(2\beta+d)}$, which matches the non-private rate up to logarithm factor (Tsybakov, 2009). The minimax bound is shown in the following theorem.

**Theorem 10.** *Denote $\mathcal{P}_{cls}$ as the set of all distributions $p$ of $\mathbf{X}$ and regression function $\eta$ that satisfy Assumption 2, $\mathcal{M}_\epsilon$ as all mechanisms satisfying $\epsilon$-LDP, then for small $\epsilon$,*

$$\inf_{\hat{Y}} \inf_{M \in \mathcal{M}_\epsilon} \sup_{(p,\eta) \in \mathcal{P}_{cls}} (R - R^*) \gtrsim (nm\epsilon^2)^{-\frac{\beta(1+\gamma)}{2(d+\beta)}} + (mn)^{-\frac{\beta(1+\gamma)}{2\beta+d}}. \tag{20}$$

The proof of Theorem 10 is shown in Appendix D.3. The comparison of Theorem 9 and Theorem 10 show that for small $\epsilon$, the upper bound and lower bound match up to a logarithmic factor. Moreover, recall (Berrett & Butucea, 2019), the minimax lower bound under item-level DP is $(N\epsilon^2)^{-\beta(1+\gamma)/(2(d+\beta))}$. If $N = nm$, this bound also matches (19), indicating that the user-level case is nearly as hard as the item-level one in asymptotic sense up to a logarithmic factor.

## 6.2 REGRESSION

For regression problem, we use the $\ell_2$ loss as the metric, i.e.

$$R = \mathbb{E}\left[(\hat{\eta}(\mathbf{X}) - \eta(\mathbf{X}))^2\right]$$

. The support is divided similarly to classification. The bounds on the convergence rate of nonparametric regression and the corresponding minimax rate are shown in the following two theorems, respectively.

**Theorem 11.** *Under Assumption 2(b) and (c), and assume that the noise is bounded, such that with probability 1, $|Y| < T$ for some $T$, if $n(\epsilon^2 \wedge 1) \geq 2c_2(\ln m + \ln n)$, in which $c_2$ is the same constant in Theorem 9, then there exists an algorithm (described in Appendix E.1), such that the risk of nonparametric regression is bounded by*

$$R \lesssim \left(\frac{mn(\epsilon^2 \wedge \epsilon)}{\ln^2 n}\right)^{-\frac{\beta}{d+\beta}} + \left(\frac{mn}{\ln n}\right)^{-\frac{2\beta}{2\beta+d}}. \tag{21}$$

**Theorem 12.** *Denote $\mathcal{P}_{reg}$ as the set of all distributions $p$ of $\mathbf{X}$ and regression function $\eta$ that satisfy the same assumption as Theorem 11, $\mathcal{Q}_\epsilon$ as all mechanisms satisfying $\epsilon$-LDP, then for small $\epsilon$,*

$$\inf_{\hat{\eta}} \inf_{Q \in \mathcal{Q}_\epsilon} \sup_{(p,\eta) \in \mathcal{P}_{reg}} R \gtrsim (nm\epsilon^2)^{-\frac{\beta}{d+\beta}} + (mn)^{-\frac{2\beta}{2\beta+d}}. \tag{22}$$

The proof of Theorem 11 and 12 are shown in Appendix E.2 and E.3, respectively. Similar to the classification, it can be found that the upper and lower bounds match up to logarithm factors.

## 7 CONCLUSION

In this paper, we have conducted a theoretical study of various statistical problems under user-level local differential privacy, including mean estimation, stochastic optimization, nonparametric classification, and regression. For each problem, we have proposed algorithms and provided information-theoretic minimax lower bounds. The results show that for many statistical problems, with the same total sample sizes, the errors under user-level and item-level $\epsilon$-LDP are nearly of the same order.

In the future, it would be interesting to relax the restriction $n(\epsilon^2 \wedge 1) \gtrsim d \ln m$. Moreover, in classification and regression problems, we assume that the pdf of $\mathbf{X}$ to be bounded away from zero. This assumption may be relaxed in our future work.

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

# A  APPENDIX

# B  ONE DIMENSIONAL MEAN ESTIMATION

## B.1  PRIVACY GUARANTEE

For $i = 1, \ldots, n/2$, the privacy mechanism is shown in step 3 in Algorithm 1. Let $\mathbf{X}'_i = \{X'_{i1}, \ldots, X'_{im}\}$ be the samples of a new user, $Z'_{ik} = \mathbf{1}(Y'_i \in B_k) + W'_{ik}$, in which $Y'_i = (\sum_{j=1}^{m} X'_{ij})/m$. The $\ell_1$ sensitivity can be bounded by $\|\mathbf{1}(Y_i \in B_k) - \mathbf{1}(Y'_i \in B_k)\|_1 \leq 2$. Therefore, it suffices to add a Laplacian noise with parameter $2/\epsilon$. For $i = n/2 + 1, \ldots, n$, the privacy mechanism is shown in step 6. Since $(R + \Delta) - (L - \Delta) = 3h + 2\Delta$, a laplacian noise with parameter $(3h + 2\Delta)/\epsilon$ suffices to guarantee user-level $\epsilon$-LDP.

## B.2 ANALYSIS OF STAGE I

In this section, we prove Lemma 2, which shows that the first stage of Algorithm 1 successes with high probability. The precise statement of this Lemma is shown as follows.

**Lemma 2.** *Let $h = 4D/\sqrt{m}$, then with probability at least $1 - \sqrt{m}e^{-c_0 n(\epsilon^2 \wedge 1)}$, $\mu \in [L, R]$, in which $c_0$ is a constant.*

Recall that for $i = 1, \ldots, n$,

$$Y_i = \frac{1}{m} \sum_{j=1}^{m} X_{ij}. \tag{23}$$

Define $p_k = P(Y \in B_k)$, in which $Y$ denotes a random variable i.i.d with $Y_1, \ldots, Y_n$. Recall that $s_k = \sum_{i=1}^{n/2} Z_{ik}$. Then we show the following lemma.

**Lemma 3.** *The following results holds. Firstly,*

$$\mathbb{E}[s_k] = \frac{1}{2} n p_k. \tag{24}$$

*Moreover, for all $t \leq n/\sqrt{2}$,*

$$P(s_k - \mathbb{E}[s_k] > t) \leq \exp\left[-\frac{1}{2\left(\frac{1}{8} + \frac{8}{\epsilon^2}\right)n} t^2\right], \tag{25}$$

*and*

$$P(s_k - \mathbb{E}[s_k] < -t) \leq \exp\left[-\frac{1}{2\left(\frac{1}{8} + \frac{8}{\epsilon^2}\right)n} t^2\right]. \tag{26}$$

*Proof.* Note that

$$\mathbb{E}[Z_{ik}] = P(Y \in B_k) = p_k, \tag{27}$$

thus

$$\mathbb{E}[s_k] = \frac{n}{2} p_k. \tag{28}$$

Now we prove (25) and (26). We first derive the sub-exponential parameters of $Z_{ik}$. Since $W_{ik}$ is Laplacian with parameter $b = 2/\epsilon$, for $|\lambda| \leq 1/(\sqrt{2}b) = \epsilon/(2\sqrt{2})$,

$$\mathbb{E}[e^{\lambda W_{ik}}] = \frac{1}{1 - b^2\lambda^2} \leq e^{2b^2\lambda^2} = e^{\frac{8}{\epsilon^2}\lambda^2}, \tag{29}$$

in which the second step uses the inequality $1/(1 - x) \leq e^{2x}$ for $x \leq 1/2$. Moreover,

$$\mathbb{E}\left[e^{\lambda(\mathbf{1}(Y_i \in B_k) - p_k)}\right] = (1 - p_k + p_k e^\lambda)e^{-\lambda p_k}. \tag{30}$$

To bound the right hand side of (30), define

$$g(\lambda) = -\lambda p_k + \ln(1 - p_k + p_k e^\lambda). \tag{31}$$

Then it can be shown that $g(0) = g'(0) = 0$, and

$$g''(\lambda) = \frac{p_k e^\lambda (1 - p_k)}{(1 - p_k + p_k e^\lambda)^2} \leq \frac{1}{4}. \tag{32}$$

Therefore, (30) can be simplified to

$$\mathbb{E}\left[e^{\lambda(\mathbf{1}(Y_i \in B_k) - p_k)}\right] \leq e^{\frac{1}{8}\lambda^2}. \tag{33}$$

From Algorithm 1, $Z_{ik} = \mathbf{1}(Y_i \in B_k) + W_{ik}$. Hence, for all $|\lambda| \leq \epsilon/(2\sqrt{2})$, from (29) and (33),

$$\mathbb{E}[e^{\lambda(Z_{ik} - \mathbb{E}[Z_{ik}])}] \leq \exp\left[\left(\frac{1}{8} + \frac{8}{\epsilon^2}\right)\lambda^2\right]. \tag{34}$$

Since $s_k = \sum_{i=1}^{n/2} Z_{ik}$, for all $|\lambda| \leq \epsilon/(2\sqrt{2})$,

$$\mathbb{E}\left[e^{\lambda(s_k - \mathbb{E}[s_k])}\right] \leq \exp\left[\frac{1}{2}\left(\frac{1}{8} + \frac{8}{\epsilon^2}\right)n\lambda^2\right], \tag{35}$$

thus if $t \leq (\epsilon/8 + 8/\epsilon)n/(2\sqrt{2})$,

$$\begin{aligned}
\mathrm{P}(s_k - \mathbb{E}[s_k] > t) &\leq \inf_{|\lambda| \leq \epsilon/(2\sqrt{2})} e^{-\lambda t} \exp\left[\frac{1}{2}\left(\frac{1}{8} + \frac{8}{\epsilon^2}\right)n\lambda^2\right] \\
&\leq \exp\left[-\frac{1}{2\left(\frac{1}{8} + \frac{8}{\epsilon^2}\right)n}t^2\right].
\end{aligned} \tag{36}$$

Similar bound holds for $\mathrm{P}(s_k - \mathbb{E}[s_k] < -t)$. Also note that $\epsilon/8 + 8/\epsilon \geq 2$. Therefore, (25) and (26) are proved for $t \leq n/\sqrt{2}$. $\qquad\square$

The next lemma bounds the values of $p_k$.

**Lemma 4.** *Denote $k^*$ as the bin index such that $\mu \in B_{k^*}$. Then*

*(1) There exists $k \in \{k^* - 1, k^*, k^* + 1\}$, $p_k \geq 1/2 - e^{-2}$;*

*(2) For all $k \notin \{k^* - 1, k^*, k^* + 1\}$, $p_k \leq 2e^{-8}$.*

*Proof.* **Proof of (1) in Lemma 4**. By Hoeffding's inequality,

$$\mathrm{P}(|Y - \mu| > t) \leq 2e^{-\frac{1}{2D^2}mt^2}, \tag{37}$$

thus

$$\mathrm{P}(|Y - \mu| \geq \frac{2D}{\sqrt{m}}) \leq 2e^{-2}. \tag{38}$$

(38) indicates that with probability at least $1 - 2e^{-2}$, $Y \in (\mu - 2D/\sqrt{m}, \mu + 2D/\sqrt{m})$. Recall that $h = 4D/\sqrt{m}$. If $\mu \geq c_{k^*}$, then $(\mu - 2D/\sqrt{m}, \mu + 2D/\sqrt{m}) \subset B_{k^*} \cup B_{k^*+1}$. Thus $p_{k^*} + p_{k^*+1} \geq 1 - 2e^{-2}$. If $\mu < c_{k^*}$, similarly, $p_{k^*} + p_{k^*-1} \geq 1 - 2e^{-2}$. Therefore, there exists a $k \in \{k^* - 1, k^*, k^* + 1\}$, such that $p_k \geq 1/2 - e^{-2}$.

**Proof of (2) in Lemma 4**. For $|k - k^*| \geq 2$,

$$\inf_{x \in B_k} |x - \mu| \geq \inf_{x \in B_k} \inf_{x' \in B_{k^*}} |x - x'| \geq h. \tag{39}$$

Therefore

$$p_k \leq \mathrm{P}(|Y - \mu| > h) = \mathrm{P}(|Y - \mu| \geq \frac{4D}{\sqrt{m}}) \leq 2e^{-8}. \tag{40}$$

$$\square$$

Based on Lemma 4, there exists $k_0 \in \{k^* - 1, k^*, k^* + 1\}$ such that $p_{k_0} \geq 1/2 - e^{-2}$. For all $k$ with $|k - k^*| \geq 2$,

$$\begin{aligned}
\mathrm{P}(\hat{k}^* = k) &\leq \mathrm{P}(s_k \geq s_{k_0}) \\
&\leq \mathrm{P}(s_k \geq n(p_k + 0.18)) + \mathrm{P}(s_{k_0} \leq n(p_{k_0} - 0.18)) \\
&\leq 2e^{-\frac{0.18^2}{2(1/8 + 8/\epsilon^2)}n} \\
&\leq 2e^{-c_0 n\epsilon^2}.
\end{aligned} \tag{41}$$

Therefore

$$\mathrm{P}(|\hat{k}^* - k^*| \geq 2) \leq 2(B-1)e^{-c_0 n\epsilon^2} \leq 2\left(\left\lceil \frac{1}{2}\sqrt{m} \right\rceil - 1\right)e^{-c_0 n\epsilon^2} \leq \sqrt{m}e^{-c_0 n\epsilon^2}, \tag{42}$$

for some constant $c_0$. Therefore, with probability at least $1 - \sqrt{m}e^{-c_0 n\epsilon^2}$, $|\hat{k}^* - k^*| \leq 1$, i.e. $\mu \in [L, R]$.

### B.3    PROOF OF THEOREM 1

In this section, we bound the mean square error of our mean estimator. Stage I has been analyzed in Section B.2. Here we focus on Stage II.

**Bound of bias.** Let

$$U = (Y \vee (L - \Delta)) \wedge (R + \Delta). \tag{43}$$

Recall that in Algorithm 1, $Z_i = (Y_i \vee (L - \Delta)) \wedge (R + \Delta) + W_i$ for $i = n/2 + 1, \dots, n$. Conditional on the first $n/2$ steps in stage I, the following relation holds:

$$\mathbb{E}[\hat{\mu}|\mathbf{Z}_{1:n/2}] = \mathbb{E}[Z_i|\mathbf{Z}_{1:n/2}] = \mathbb{E}[U|\mathbf{Z}_{1:n/2}]. \tag{44}$$

To bound the bias of $\hat{\mu}$, it suffices to bound $|\mathbb{E}[U] - \mu|$. From (43),

$$\begin{aligned}
\mathbb{E}[U|\mathbf{Z}_{1:n/2}] &= \mathbb{E}[Y\mathbf{1}(L - \Delta \leq Y \leq R + \Delta)|\mathbf{Z}_{1:n/2}] \\
&\quad + (L - \Delta)\mathrm{P}(Y < L - \Delta|\mathbf{Z}_{1:n/2}) + (R + \Delta)\mathrm{P}(Y > R + \Delta|\mathbf{Z}_{1:n/2}). 
\end{aligned} \tag{45}$$

Moreover,

$$\begin{aligned}
\mu &= \mathbb{E}[Y] \\
&= \mathbb{E}[Y\mathbf{1}(L - \Delta \leq Y \leq R + \Delta)] + \mathbb{E}[Y\mathbf{1}(Y < L - \Delta)] + \mathbb{E}[Y\mathbf{1}(Y > R + \Delta)]. 
\end{aligned} \tag{46}$$

Note that

$$\begin{aligned}
&\mathbb{E}[Y\mathbf{1}(Y > R + \Delta)|\mathbf{Z}_{1:n/2}] \\
&= \mathbb{E}[(Y - R - \Delta)\mathbf{1}(Y > R + \Delta)|\mathbf{Z}_{1:n/2}] + (R + \Delta)\mathrm{P}(Y > R + \Delta|\mathbf{Z}_{1:n/2}) \\
&= \int_0^\infty \mathrm{P}(Y > R + \Delta + t|\mathbf{Z}_{1:n/2})dt + (R + \Delta)\mathrm{P}(Y > R + \Delta|\mathbf{Z}_{1:n/2}), 
\end{aligned} \tag{47}$$

and similarly,

$$\begin{aligned}
&\mathbb{E}[Y\mathbf{1}(Y < L - \Delta)|\mathbf{Z}_{1:n/2}] \\
&= -\mathbb{E}[(L - \Delta - Y)\mathbf{1}(Y < L - \Delta)|\mathbf{Z}_{1:n/2}] + (L - \Delta)\mathrm{P}(Y < L - \Delta|\mathbf{Z}_{1:n/2}) \\
&= (L - \Delta)\mathrm{P}(Y < L - \Delta|\mathbf{Z}_{1:n/2}) - \int_0^\infty \mathrm{P}(Y < L - \Delta - t|\mathbf{Z}_{1:n/2})dt. 
\end{aligned} \tag{48}$$

From (45), (46), (47) and (48), the bias of $\hat{\mu}$ can be bounded by

$$|\mathbb{E}[U] - \mu| = \left| \int_0^\infty \mathrm{P}(Y > R + \Delta + t)dt - \int_0^\infty \mathrm{P}(Y < L - \Delta - t)dt \right|. \tag{49}$$

Denote $E_1$ as the event that stage I is successful, i.e. $\mu \in [L, R]$. Conditional on $E_1$,

$$\begin{aligned}
\int_0^\infty \mathrm{P}(Y > R + \Delta + t|E_1)dt &\leq \int_0^\infty \mathrm{P}(|Y - \mu| > R + \Delta - \mu + t|E_1)dt \\
&\leq \int_\Delta^\infty \mathrm{P}(|Y - \mu| > t)dt \\
&\overset{(a)}{\leq} 2\int_\Delta^\infty e^{-\frac{m}{2D^2}t^2}dt \\
&= \frac{2D}{\sqrt{m}} \int_{\sqrt{m}\Delta/D}^\infty e^{-\frac{1}{2}u^2}du \\
&\overset{(b)}{\leq} \frac{2\sqrt{2\pi}D}{\sqrt{m}} e^{-\frac{1}{2}\left(\frac{\sqrt{m}\Delta}{D}\right)^2} \\
&\overset{(c)}{=} \frac{2\sqrt{2\pi}D}{\sqrt{mn}}. 
\end{aligned} \tag{50}$$

(a) uses Hoeffding's inequality. (b) uses the inequality $\int_s^\infty e^{-\frac{1}{2}u^2}du \leq \sqrt{2\pi}e^{-\frac{1}{2}s^2}$. For (c), recall that $\Delta = D\sqrt{\ln n/m}$. Similarly,

$$\int_0^\infty \mathrm{P}(Y < L - \Delta - t|E_1)dt \leq \frac{2\sqrt{2\pi}D}{\sqrt{mn}}. \tag{51}$$

Therefore, from (44), (49), (51) and (50), under $E_1$,

$$|\mathbb{E}[\hat{\mu}|\mathbf{Z}_{1:n/2}] - \mu| \leq \frac{4\sqrt{2\pi}D}{\sqrt{mn}}. \tag{52}$$

If $E_1$ is not satisfied, then $|\hat{\mu} - \mu| \leq 2D$. Hence

$$|\mathbb{E}[\hat{\mu}] - \mu| = |\mathbb{E}[U] - \mu| \leq \frac{4\sqrt{2\pi}D}{\sqrt{mn}} + 2D\mathrm{P}(E_1^c), \tag{53}$$

**Bound of Variance.** Let $\mathrm{Var}[X] := \sigma^2$. Since $X \in [-D, D]$, $\sigma^2 \leq D^2$ holds. Therefore

$$\mathrm{Var}[Z_i] \leq \mathrm{Var}[Y] + \mathrm{Var}[W_i] = \frac{\sigma^2}{m} + 2\frac{(3h + 2\Delta)^2}{\epsilon^2}. \tag{54}$$

Thus

$$\mathrm{Var}[\hat{\mu}] \leq \frac{\sigma^2}{mn} + \frac{2(3h + 2\Delta)^2}{n\epsilon^2}. \tag{55}$$

Recall that $h = 4D/\sqrt{m}$, $\Delta = D\sqrt{\ln n/m}$, $\mathrm{P}(E_1^c) \leq \sqrt{m}e^{-c_0 n\epsilon^2}$, the mean squared error can be bounded by

$$\mathbb{E}[(\hat{\mu} - \mu)^2] \lesssim \frac{D^2 \ln n}{nm\epsilon^2} + \frac{D^2}{mn}. \tag{56}$$

### B.4 PROOF OF THEOREM 2

Let $V$ be a random variable taking values in $\{-1, 1\}$ with equal probability. Construct the distribution of $X$ as following:

$$\mathrm{P}(X = D|V = v) = \frac{1 + sv}{2}, \mathrm{P}(X = -D) = \frac{1 - sv}{2}, \tag{57}$$

in which $0 < s \leq 1/2$. Define

$$\mu_+ = \mathbb{E}[X|V = 1], \tag{58}$$
$$\mu_- = \mathbb{E}[X|V = -1], \tag{59}$$

then $\mu_+ = Ds$, $\mu_- = -Ds$.

Denote

$$\hat{V} = \mathbf{1}(\hat{\mu} > 0). \tag{60}$$

Then

$$\mathbb{E}[(\hat{\mu} - \mu)^2] \geq D^2 s^2 \mathrm{P}(\hat{V} \neq V). \tag{61}$$

Given $X_{ij}$, $i = 1, \ldots, n$, $j = 1, \ldots, m$, by a private mechanism, we observe $\mathbf{Z}_i$, $i = 1, \ldots, n$. Denote $p_+$ and $p_-$ as the distribution of $\mathbf{Z}_i$ conditional on $V = 1$ and $V = -1$, respectively. Correspondingly, let $p_+^n$ and $p_-^n$ be the joint distribution of $\mathbf{Z}_1, \ldots, \mathbf{Z}_n$. $p_{X+}$ and $p_{X-}$ denotes the distribution of $X_{ij}$ under $V = 1$ and $V = -1$, respectively. $p_{X+}^m$ and $p_{X-}^m$ are the corresponding joint distribution of $X_{i1}, \ldots, X_{im}$, i.e. all samples of a user. Then

$$\mathrm{P}(\hat{V} \neq V) \overset{(a)}{\geq} \frac{1}{2}\left(1 - \mathbb{TV}(p_+^n, p_-^n)\right)$$

$$\overset{(b)}{\geq} \frac{1}{2}\left(1 - \sqrt{\frac{1}{2}D_{KL}(p_+^n || p_-^n)}\right)$$

$$\overset{(c)}{\geq} \frac{1}{2}\left(1 - \sqrt{\frac{1}{2}nD_{KL}(p_+ || p_-)}\right)$$

$$\overset{(d)}{\geq} \frac{1}{2}\left(1 - \sqrt{\frac{1}{2}n(e^\epsilon - 1)^2 \mathbb{TV}^2(p_{X+}^m, p_{X-}^m)}\right)$$

$$\overset{(e)}{\geq} \frac{1}{2}\left(1 - \frac{1}{2}\sqrt{nm(e^\epsilon - 1)^2 D_{KL}(p_{X+} || p_{X-})}\right). \tag{62}$$

In (a), $\mathbb{TV}$ is the total variation distance. (b) uses Pinsker's inequality, and $D_{KL}$ denotes the Kullback-Leibler (KL) divergence. (c) uses the property of KL divergence. (d) comes from Theorem 1 in (Duchi et al., 2018). Finally, (e) uses Pinsker's inequality again.

From (57),

$$D(p_{X+}||p_{X-}) = \frac{1+s}{2}\ln\frac{1+s}{1-s} + \frac{1-s}{2}\ln\frac{1-s}{1+s} = s\ln\frac{1+s}{1-s} \leq 3s^2, \tag{63}$$

in which the last step holds because $0 < s < 1/2$.

With $\epsilon < 1$, let $s \sim 1/\sqrt{nm\epsilon^2}$, then $\mathrm{P}(\hat{V} \neq V) \sim 1$. Hence

$$\inf_{\hat{\mu}} \inf_{Q \in \mathcal{Q}_\epsilon} \sup_{p \in \mathcal{P}_\mathcal{X}} \mathbb{E}[(\hat{\mu} - \mu)^2] \gtrsim \frac{D^2}{nm\epsilon^2}. \tag{64}$$

If $\epsilon > 1$, then from standard minimax analysis for non-private problems, the estimation error can not be smaller than $\sigma^2/(mn)$, with $\sigma^2$ being the sample variance. The maximum value of $\sigma^2$ is $D^2$. Therefore it can be easily shown that

$$\inf_{\hat{\mu}} \inf_{Q \in \mathcal{Q}_\epsilon} \sup_{p \in \mathcal{P}_\mathcal{X}} \mathbb{E}[(\hat{\mu} - \mu)^2] \gtrsim \frac{D^2}{nm}. \tag{65}$$

**Limit of using fixed number of users.** Finally, we prove the results for fixed $n$, which shows that zero error can not be reached even with $m \to \infty$. Recall that $p_+$ and $p_-$ are the distribution of $\mathbf{Z}_i$ conditional on $V = 1$ and $V = -1$. $\mathbf{Z}_i$ is $\epsilon$-DP with respect to $\mathbf{X}_{i1}, \ldots, \mathbf{X}_{im}$, thus $|\ln p_+(S)/p_-(S)| \leq \epsilon$ for all set $S$, and then it can be shown that $D_{KL}(p_+||p_-) \leq \epsilon(e^\epsilon - 1)$ (Dwork et al., 2014). Therefore

$$
\begin{aligned}
\mathrm{P}(\hat{V} \neq V) &\geq \frac{1}{2}(1 - \mathbb{TV}(p_+^n, p_-^n)) \\
&\geq \frac{1}{4}e^{-D_{KL}(p_+^n||p_-^n} \\
&= \frac{1}{4}e^{-nD_{KL}(p_+||p_-)} \\
&\geq \frac{1}{4}e^{-n\epsilon(e^\epsilon-1)}.
\end{aligned} \tag{66}
$$

Let $s = 1$ in (61), then

$$\mathbb{E}[(\hat{\mu} - \mu)^2] \geq \frac{1}{4}D^2 e^{-n\epsilon(e^\epsilon-1)}. \tag{67}$$

### B.5 PROOF OF THEOREM 3

For unbounded support, the user-wise average values are clipped to $[-D, D]$, i.e.

$$Y_i = -D \vee \left(\frac{1}{m}\sum_{j=1}^m X_{ij} \wedge D\right), \tag{68}$$

which means to clip the average value of each user to $[-D, D]$. Now for simplicity, let $Y$ be a random variable i.i.d with $Y_i, i = 1, \ldots, n$. Define

$$\mu_T := \mathbb{E}[Y]. \tag{69}$$

Recall that in Algorithm 1, $Z_i = (Y_i \vee (L - \Delta)) \wedge (R + \Delta) + W_i$ and $\hat{\mu} = (2/n)\sum_{i=n/2+1}^n Z_i$. Thus

$$\mathbb{E}[\hat{\mu}|\mathbf{Z}_{1:n/2}] = \mathbb{E}[Z_i|\mathbf{Z}_{1:n/2}] = \mathbb{E}[U|\mathbf{Z}_{1:n/2}]. \tag{70}$$

The bias of $\hat{\mu}$ can be bounded by

$$|\mathbb{E}[\hat{\mu}|\mathbf{Z}_{1:n/2}] - \mu| \leq |\mathbb{E}[U|\mathbf{Z}_{1:n/2}] - \mu_T| - |\mu_T - \mu|. \tag{71}$$

Now we bound two terms in the right hand side of (71) separately.

**Bound of** $|\mathbb{E}[U] - \mu_T|$**.**

Similar to (49), following steps (45), (46), (47) and (48), it can be shown that

$$|\mathbb{E}[U|\mathbf{Z}_{1:n/2}] - \mu_T| = \left| \int_0^\infty \mathrm{P}(Y > R + \Delta + t)dt - \int_0^\infty \mathrm{P}(Y < L - \Delta - t)dt \right|. \tag{72}$$

Denote $E_1$ as the event that stage I is successful, i.e. $\mu \in [L, R]$. To bound the right hand side of (72), we use the following Lemma.

**Lemma 5.** *(Restated from Corollary 6 in (Bakhshizadeh et al., 2023)) If $X_1, \ldots, X_m$ are $m$ i.i.d copies of random variable $X$ with $\mathbb{E}[|X|^p] \leq M_p < \infty$, $m \geq 2$, then for any constant $c$, there exists a constant $C$, such that for all $t \geq cM_p^{1/p}\sqrt{\ln m}$,*

$$P\left( \left| \frac{1}{m} \sum_{j=1}^m X_j - \mu \right| > t\sqrt{\frac{1}{m}} \right) \leq CM_p t^{-p} m^{-\left(\frac{p}{2}-1\right)}. \tag{73}$$

According to Lemma 5, with

$$\Delta \geq cM_p^{1/p}\sqrt{\ln m/m}, \tag{74}$$

the following bound holds:

$$
\begin{aligned}
\int_0^\infty \mathrm{P}(Y > R + \Delta + t|E_1)dt &\leq \int_\Delta^\infty \mathrm{P}(|Y - \mu| > t)dt \\
&\leq \int_\Delta^\infty CM_p t^{-p} m^{-(p-1)}dt \\
&\leq \frac{CM_p}{p-1} m^{-(p-1)} \Delta^{-(p-1)}.
\end{aligned}
\tag{75}
$$

Therefore from (49),

$$|\mathbb{E}[\hat{\mu}] - \mu_T| \leq \frac{2CM_p}{p-1} m^{-(p-1)} \Delta^{-(p-1)} + 2DP(E_1^c), \tag{76}$$

Similar to Lemma 2, it can be shown that $\mathrm{P}(E_1^c)$ decays exponentially to zero if $D \lesssim e^{c_2 n \epsilon^2}$ for some constant $c_2$.

**Bound of** $|\mu_T - \mu|$**.** Denote $\bar{X}$ as a random variable i.i.d with $(1/m)\sum_{j=1}^m X_{ij}$, and $Y$ can be viewed as $\bar{X}$ clipped by $[-D, D]$, i.e. $Y = -D \vee (\bar{X} \wedge D)$. Then

$$\mu = \mathbb{E}[\bar{X}\mathbf{1}(-D \leq \bar{X} \leq D)] + \mathbb{E}[\bar{X}\mathbf{1}(\bar{X} > D)] + \mathbb{E}[\bar{X}\mathbf{1}(\bar{X} < -D)], \tag{77}$$

$$\mu_T = \mathbb{E}[\bar{X}\mathbf{1}(-D \leq \bar{X} \leq D)] + D\mathrm{P}(\bar{X} > D) - D\mathrm{P}(\bar{X} < -D). \tag{78}$$

For sufficiently large $m, n$, $D > \mu/2$ holds, thus

$$
\begin{aligned}
\mathbb{E}[\bar{X}\mathbf{1}(\bar{X} > D)] - D\mathrm{P}(\bar{X} > D) &= \int_D^\infty \mathrm{P}(\bar{X} > t)dt \\
&\leq \int_D^\infty \mathrm{P}(\bar{X} - \mu > \frac{t}{2})dt \\
&\leq \int_D^\infty 2^p CM_p m^{-(p-1)} t^{-p}dt \\
&\lesssim M_p m^{-(p-1)} D^{-(p-1)},
\end{aligned}
\tag{79}
$$

in which the third step uses Lemma 5. Hence

$$|\mu_T - \mu| \lesssim M_p m^{-(p-1)} D^{-(p-1)}. \tag{80}$$

Hence from (71), the bias can be bounded by

$$|\mathbb{E}[\hat{\mu}] - \mu| \lesssim M_p m^{-(p-1)} \Delta^{-(p-1)} + M_p D^{-(p-1)} m^{-(p-1)}. \tag{81}$$

For the variance of $\hat{\mu}$, (55) still holds, i.e.

$$\text{Var}[\hat{\mu}] \leq \frac{\sigma^2}{mn} + \frac{2(3h+2\Delta)^2}{n\epsilon^2} \lesssim \frac{M_p^{2/p}}{mn} + \frac{\Delta^2}{n\epsilon^2}, \tag{82}$$

in which the variance is bounded using Hölder inequality. From (81) and (82), the mean squared error can be bounded by

$$\mathbb{E}[(\hat{\mu} - \mu_T)^2] \lesssim M_p^2 m^{-2(p-1)} \Delta^{-2(p-1)} + M_p^2 D^{-2(p-1)} m^{-2(p-1)} + \frac{\Delta^2}{n\epsilon^2} + \frac{M_p^{2/p}}{mn}. \tag{83}$$

We pick $\delta$ to minimize the right hand side of (83). Meanwhile, the restriction (74) also needs to be guaranteed. Therefore, let

$$\Delta = c M_p^{1/p} \sqrt{\frac{\ln m}{m}} \vee (M_p^2 n\epsilon^2)^{\frac{1}{2p}} m^{-\left(1-\frac{1}{p}\right)}. \tag{84}$$

Then

$$\mathbb{E}[(\hat{\mu} - \mu)^2] \lesssim M_p^{2/p} \left[ \frac{\ln m}{mn\epsilon^2} \vee (M_p m^2 n\epsilon^2)^{-\left(1-\frac{1}{p}\right)} + D^{-2(p-1)} m^{-2(p-1)} + \frac{1}{mn} \right]. \tag{85}$$

If $D \gtrsim \Delta$, then the second term in (85) will not dominate. Now the proof of Theorem 3 is complete. Recall that $D \lesssim e^{c_2 n\epsilon^2}$ is needed to ensure that stage $I$ success with high probability, the suitable range of $D$ is

$$\Delta \lesssim D \lesssim e^{c_2 n\epsilon^2}. \tag{86}$$

## C  MULTI-DIMENSIONAL MEAN ESTIMATION

### C.1  PROOF OF THEOREM 5

Transformation with Kashin's representation $\mathbf{X}' = \mathbf{U}\mathbf{X}$ converts $\ell_2$ support to $\ell_\infty$ support. The only difference is that now the supremum norm reduces from $D$ to $KD/\sqrt{d}$. Hence, from Theorem 4,

$$\mathbb{E}\left[\left\|\hat{\theta} - \theta\right\|_2^2\right] \lesssim \frac{D^2}{nm}\left(1 + \frac{d\ln n}{\epsilon^2 \wedge \epsilon}\right). \tag{87}$$

Recall that the final estimator is $\hat{\mu} = \mathbf{U}^T \hat{\theta}$. Moreover, by Lemma 1, $\mathbf{U}^T\mathbf{U} = \mathbf{I}_d$. Define $v = \hat{\theta} - \mathbf{U}\mu$, then $\mathbf{U}^T\mathbf{v} = 0$. Therefore

$$
\begin{aligned}
\mathbb{E}\left[\left\|\hat{\theta} - \theta\right\|_2^2\right] &\overset{(a)}{=} \mathbb{E}\left[\|\mathbf{U}\hat{\mu} + \mathbf{v} - \mathbf{U}\mu\|_2^2\right] \\
&= \mathbb{E}\left[\|\mathbf{U}(\hat{\mu} - \mu)\|_2^2\right] + \mathbb{E}[\|\mathbf{v}\|^2] + 2\mathbb{E}\left[(\hat{\mu} - \mu)^T\mathbf{U}^T\mathbf{v}\right] \\
&= \mathbb{E}\left[\|\mathbf{U}(\hat{\mu} - \mu)\|_2^2\right] + \mathbb{E}[\|\mathbf{v}\|^2] \\
&\geq \mathbb{E}\left[\|\mathbf{U}(\hat{\mu} - \mu)\|_2^2\right] \\
&= \mathbb{E}\left[\mathbf{U}(\hat{\mu} - \mu)(\hat{\mu} - \mu)^T\mathbf{U}^T\right] \\
&= \mathbb{E}\left[\text{tr}((\hat{\mu} - \mu)(\hat{\mu} - \mu)^T\mathbf{U}^T\mathbf{U})\right] \\
&\overset{(b)}{=} \mathbb{E}\left[\text{tr}((\hat{\mu} - \mu)(\hat{\mu} - \mu)^T)\right] \\
&= \mathbb{E}[\|\hat{\mu} - \mu\|_2^2].
\end{aligned}
\tag{88}
$$

From (87),

$$\mathbb{E}[\|\hat{\mu} - \mu\|_2^2] \lesssim \frac{D^2}{nm}\left(1 + \frac{d\ln n}{\epsilon^2 \wedge \epsilon}\right), \tag{89}$$

in which (a) holds since $\theta = \mathbf{U}\mu$, and (b) uses Lemma 1.

## C.2  PROOF OF THEOREM 6

Denote $\mathcal{V} = \{-1, 1\}^d$. For $\mathbf{v} \in \mathcal{V}$, let

$$P(\mathbf{X} = D\mathbf{e}_k) = \frac{1 + sv_k}{2d}, \tag{90}$$

$$P(\mathbf{X} = -D\mathbf{e}_k) = \frac{1 - sv_k}{2d}, \tag{91}$$

for $k = 1, \ldots, d$, in which $\mathbf{e}_k$ is the unit vector towards $k$-th coordinate, $0 < s \le 1/2$, and $v_k$ is the $k$-th element of $\mathbf{v}$. Denote $\mu_k = \mathbb{E}[\mathbf{X} \cdot \mathbf{e}_k]$ as the $k$-th component of $\mu$. Then

$$\mu_k = D\frac{1 + sv_k}{2d} - D\frac{1 - sv_k}{2d} = \frac{D}{d} sv_k. \tag{92}$$

Let $\hat{\mu}_k$ be the $k$-th component of $\hat{\mu}$, and

$$\hat{v}_k = \mathbf{1}(\hat{\mu}_k > 0). \tag{93}$$

If $\hat{v}_k \ne v_k$, then $|\hat{\mu}_k - \mu_k| \ge Ds/d$. Hence

$$\mathbb{E}\left[\|\hat{\mu} - \mu\|_2^2\right] = \mathbb{E}\left[\sum_{k=1}^d (\hat{\mu}_k - \mu_k)^2\right] \ge \frac{D^2}{d^2} s^2 \mathbb{E}[\rho_H(\hat{\mathbf{v}}, \mathbf{v})], \tag{94}$$

in which

$$\rho_H(\hat{\mathbf{v}}, \mathbf{v}) = \sum_{k=1}^d \mathbf{1}(\hat{v}_k \ne v_k) \tag{95}$$

is the Hamming distance. Therefore the minimax lower bound can be transformed to the following form:

$$\inf_{\hat{\mu}} \inf_{Q \in \mathcal{Q}_\epsilon} \sup_{p \in \mathcal{P}_{\mathcal{X}, 1}} \mathbb{E}\left[\|\hat{\mu} - \mu\|_2^2\right] \ge \frac{D^2}{d^2} s^2 \inf_{\hat{\mathbf{v}}} \inf_{Q \in \mathcal{Q}_\epsilon} \sup_{\mathbf{v} \in \mathcal{V}} \mathbb{E}[\rho_H(\hat{\mathbf{v}}, \mathbf{v})]. \tag{96}$$

Define

$$\delta = \sup_{Q \in \mathcal{Q}_\epsilon} \max_{\mathbf{v}, \mathbf{v}': \rho_H(\mathbf{v}, \mathbf{v}')=1} D(p_{\mathbf{Z}|\mathbf{v}} || p_{\mathbf{Z}|\mathbf{v}'}), \tag{97}$$

in which $p_{\mathbf{Z}|\mathbf{v}}$ is the distribution of $\mathbf{Z}_i$ when $\mathbf{X}_{i1}, \ldots, \mathbf{X}_{im}$ are distributed according to (90) and (91). By Theorem 2.12 (iv) in (Tsybakov, 2009),

$$\inf_{\hat{\mathbf{v}}} \inf_{Q \in \mathcal{Q}_\epsilon} \sup_{\mathbf{v} \in \mathcal{V}} \mathbb{E}[\rho_H(\hat{\mathbf{v}}, \mathbf{v})] \ge \frac{d}{2} \max\left(\frac{1}{2} e^{-\delta}, 1 - \sqrt{\frac{\delta}{2}}\right). \tag{98}$$

Now it remains to bound $\beta$. From Theorem 1 in (Duchi et al., 2018),

$$D(p_{\mathbf{Z}|\mathbf{v}} || p_{\mathbf{Z}|\mathbf{v}'}) \le n(e^\epsilon - 1)^2 \mathbb{TV}^2(p_{\mathbf{X}|\mathbf{v}}^m, p_{\mathbf{X}|\mathbf{v}'}^m). \tag{99}$$

To bound the total variation distance, we use a generalized version of Pinsker's inequality, stated in Lemma 10. Without loss of generality, suppose $\mathbf{v}, \mathbf{v}'$ is different at the first component. Then

$$
\begin{aligned}
\mathbb{TV}^2(p_{\mathbf{X}|\mathbf{v}}^m, p_{\mathbf{X}|\mathbf{v}'}^m) &\le \frac{1}{2} p_{\mathbf{X}|\mathbf{v}}(\{D\mathbf{e}_1, -D\mathbf{e}_1\}) D(p_{\mathbf{X}|\mathbf{v}}^m || p_{\mathbf{X}|\mathbf{v}'}^m) \\
&= \frac{1}{2d} D(p_{\mathbf{X}|\mathbf{v}}^m || p_{\mathbf{X}|\mathbf{v}'}^m) \\
&= \frac{m}{2d} D(p_{\mathbf{X}|\mathbf{v}} || p_{\mathbf{X}|\mathbf{v}'}) \\
&= \frac{m}{2d} \left(\frac{1 + s}{2d} \ln \frac{1 + s}{1 - s} + \frac{1 - s}{2d} \ln \frac{1 - s}{1 + s}\right) \\
&= \frac{m}{2d} \frac{s}{d} \ln \frac{1 + s}{1 - s} \\
&\le \frac{3ms^2}{2d^2}, 
\end{aligned}
\tag{100}
$$

in which the last step holds since $0 < s \leq 1/2$. Therefore

$$\delta \leq \frac{3}{2}n(e^\epsilon - 1)^2 \frac{ms^2}{d^2}. \tag{101}$$

To ensure $\delta \lesssim 1$, let

$$s \sim \frac{d}{\sqrt{mn\epsilon^2}} \wedge 1, \tag{102}$$

then

$$\inf_{\hat{\mathbf{v}}} \inf_{Q \in \mathcal{Q}_\epsilon} \sup_{\mathbf{v} \in \mathcal{V}} \mathbb{E}[\rho_H(\hat{\mathbf{v}}, \mathbf{v})] \gtrsim d. \tag{103}$$

Hence

$$\inf_{\hat{\mu}} \inf_{Q \in \mathcal{Q}_\epsilon} \sup_{p \in \mathcal{P}_{\mathcal{X},1}} \mathbb{E}\left[\|\hat{\mu} - \mu\|_2^2\right] \gtrsim \frac{D^2}{d}s^2 \sim \frac{D^2}{d}\left(\frac{d^2}{mn\epsilon^2} \wedge 1\right) \sim \frac{D^2 d}{mn\epsilon^2} \wedge \frac{D^2}{d}. \tag{104}$$

Moreover, from standard minimax analysis for non-private problems (Tsybakov, 2009), it can be shown that

$$\inf_{\hat{\mu}} \inf_{Q \in \mathcal{Q}_\epsilon} \sup_{p \in \mathcal{P}_{\mathcal{X},1}} \mathbb{E}\left[\|\hat{\mu} - \mu\|_2^2\right] \gtrsim \frac{D^2}{mn}. \tag{105}$$

### C.3 PROOF OF THEOREM 7

Without loss of generality, suppose $d$ is a power of 2, which enables the construction of a Hadamard matrix $\mathbf{H}_d = (\mathbf{h}_1, \ldots, \mathbf{h}_d)$ by Sylvesters' approach (Yarlagadda & Hershey, 2012). Then $\mathbf{h}_k^T \mathbf{h}_l = 0$, $\forall k \neq l$ and $h_k^T h_k = d$. Denote $\mathcal{V} = \{-1, 1\}^d$. For $\mathbf{v} \in \mathcal{V}$, let

$$\mathbf{P}(\mathbf{X} = D\mathbf{h}_k) = \frac{1 + sv_k}{2d}, \tag{106}$$

$$\mathbf{P}(\mathbf{X} = -D\mathbf{h}_k) = \frac{1 - sv_k}{2d}, \tag{107}$$

for $k = 1, \ldots, d$, $s \in (0, 1/2]$. Then

$$\mathbf{h}_k^T \mu_k = \mathbb{E}[\mathbf{h}_k^T \mathbf{X}] = D\mathbf{h}_k^T \mathbf{h}_k \frac{1 + sv_k}{2d} - D\mathbf{h}_k^T \mathbf{h}_k \frac{1 - sv_k}{2d} = Dsv_k. \tag{108}$$

Let

$$\hat{v}_k = \mathbf{1}(\mathbf{h}_k^T \hat{\mu}_k > 0). \tag{109}$$

If $\hat{v}_k \neq v_k$, then $|\mathbf{h}_k^T(\hat{\mu}_k - \mu_k)| > Ds$. Hence

$$\begin{aligned}
\mathbb{E}\left[\|\hat{\mu} - \mu\|_2^2\right] &= \frac{1}{d}\mathbb{E}[(\hat{\mu} - \mu)^T \mathbf{H}_d \mathbf{H}_d^T (\hat{\mu} - \mu)] \\
&= \frac{1}{d}\mathbb{E}\left[\sum_{k=1}^{d}(\mathbf{h}_k^T(\hat{\mu}_k - \mu_k))^2\right] \\
&\geq \frac{D^2}{d}s^2 \mathbb{E}[\rho_H(\hat{\mathbf{v}}, \mathbf{v})].
\end{aligned} \tag{110}$$

The result is $d$ times larger than (94). The remaining steps just follow the case with $\ell_1$ support, i.e. Section C.2. The result is

$$\inf_{\hat{\mu}} \inf_{Q \in \mathcal{Q}_\epsilon} \sup_{p \in \mathcal{P}_{\mathcal{X},\infty}} \mathbb{E}\left[\|\hat{\mu} - \mu\|_2^2\right] \gtrsim \frac{D^2 d^2}{mn(e^\epsilon - 1)^2} + \frac{D^2}{mn}. \tag{111}$$

### C.4 HIGH DIMENSIONAL MEAN ESTIMATION WITH HEAVY TAILS

We start from the case that $\mathbb{E}[|X_k|^p] \leq M_p$ for all $k$. Then follow steps from (5) to (7), using Theorem 3, the following bounds can be obtained immmediately.

If $\epsilon < 1$, then

$$\mathbb{E}[(\hat{\mu}_k - \mu_k)^2] \lesssim M_p^{2/p} \left[ \frac{d \ln m}{mn\epsilon^2} \vee \left( \frac{m^2 n \epsilon^2}{d} \right)^{1-1/p} + \frac{d}{mn} \right]. \tag{112}$$

If $1 \leq \epsilon < d \ln m$, then

$$\mathbb{E}[(\hat{\mu}_k - \mu_k)^2] \lesssim M_p^{2/p} \left[ \frac{d \ln m}{mn\epsilon} \vee \left( \frac{m^2 n \epsilon}{d} \right)^{-(1-1/p)} + \frac{d}{mn\epsilon} \right]. \tag{113}$$

Finally, if $\epsilon \geq d \ln m$, then

$$\mathbb{E}[(\hat{\mu}_k - \mu_k)^2] \lesssim M_p^{2/p} \left[ \frac{d^2 \ln m}{mn\epsilon^2} \vee \left( \frac{m^2 n \epsilon^2}{d} \right)^{1-1/p} + \frac{1}{mn} \right]. \tag{114}$$

Combine all these three cases, we get

$$\mathbb{E}[(\hat{\mu}_k - \mu_k)^2] \lesssim M_p^{2/p} \left[ \frac{d \ln m}{mn(\epsilon^2 \wedge \epsilon)} \vee \left( \frac{d}{m^2 n(\epsilon^2 \wedge \epsilon)} \right)^{1-1/p} + \frac{1}{mn} \right]. \tag{115}$$

Therefore

$$\mathbb{E}\left[ \|\hat{\mu} - \mu\|_2^2 \right] \lesssim M_p^{2/p} \left[ \frac{d^2 \ln m}{mn(\epsilon^2 \wedge \epsilon)} \vee \left( \frac{d}{m^2 n(\epsilon^2 \wedge \epsilon)} \right)^{1-1/p} + \frac{d}{mn} \right]. \tag{116}$$

Now move on to the case with $\mathbb{E}[\|\mathbf{X}\|_2^p] \leq M_p$. Then we still conduct transformation using Kashin's representation. By Lemma 1,

$$\|\mathbf{Ux}\|_\infty \leq \frac{K}{\sqrt{d}} \|\mathbf{x}\|_2. \tag{117}$$

Thus

$$\begin{aligned} \mathbb{E}[\|\mathbf{UX}\|_\infty^p] &\leq \frac{K^p}{d^{p/2}} \mathbb{E}[\|\mathbf{X}\|_2^p] \\ &\leq K^p M_p d^{-p/2}. \end{aligned} \tag{118}$$

Therefore, for each unit vector $\mathbf{e}_k$ for the $k$-th coordinate,

$$\mathbb{E}[|\mathbf{e}_k^T \mathbf{UX}|^p] \leq K^p M_p d^{-p/2}. \tag{119}$$

Let $\theta = \mathbf{U}\mu$, and then estimate $\theta$ using $\mathbf{UX}_{ij}$, $i = 1, \ldots, n$, $j = 1, \ldots, m$. Then we replace $M_p$ in (116) with $K^p M_p d^{-p/2}$. Therefore

$$\mathbb{E}\left[ \left\| \hat{\theta} - \theta \right\|_2^2 \right] \lesssim M_p^{2/p} \left[ \frac{d \ln m}{mn(\epsilon^2 \wedge \epsilon)} \vee \left( \frac{d}{m^2 n(\epsilon^2 \wedge \epsilon)} \right)^{1-1/p} + \frac{1}{mn} \right]. \tag{120}$$

From (88),

$$\mathbb{E}\left[ \|\hat{\mu} - \mu\|_2^2 \right] \lesssim M_p^{2/p} \left[ \frac{d \ln m}{mn(\epsilon^2 \wedge \epsilon)} \vee \left( \frac{d}{m^2 n(\epsilon^2 \wedge \epsilon)} \right)^{1-1/p} + \frac{1}{mn} \right]. \tag{121}$$

## D    STOCHASTIC OPTIMIZATION

This section proves Theorem 8. From Theorem 5, we have

$$\mathbb{E}\left[\|g_t - \nabla L(\theta_t)\|_2^2\right] \leq \frac{CD^2T}{nm}\left(1 + \frac{d\ln n}{\epsilon^2 \wedge \epsilon}\right) \tag{122}$$

for some constant $C$. Recall that the update rule is

$$\theta_{t+1} = \theta_t - \eta\mathbf{g}_t. \tag{123}$$

Then

$$
\begin{aligned}
\|\theta_{t+1} - \theta^*\|_2 &= \|\theta_t - \eta\mathbf{g}_t - \theta^*\|_2 \\
&\leq \|\theta_t - \eta\nabla L(\theta_t) - \theta^*\|_2 + \eta\|\nabla L(\theta_t) - \mathbf{g}_t\|_2 .
\end{aligned} \tag{124}
$$

The first term can be bounded by

$$
\begin{aligned}
&\|\theta_t - \eta\nabla L(\theta_t) - \theta^*\|_2^2 \\
&= \|\theta_t - \theta^*\|_2^2 - 2\eta\langle\theta_t - \theta^*, \nabla L(\theta_t)\rangle + \eta^2\|\nabla L(\theta_t)\|_2^2 \\
&\overset{(a)}{\leq} \|\theta_t - \theta^*\|_2^2 - 2\eta\left(L(\theta_t) - L(\theta^*) + \frac{\gamma}{2}\|\theta_t - \theta^*\|_2^2\right) + \eta^2\|\nabla L(\theta_t)\|_2^2 \\
&\overset{(b)}{\leq} (1 - \eta\gamma)\|\theta_t - \theta^*\|_2^2 - 2\eta(L(\theta_t) - L(\theta^*)) + 2\eta^2 G(L(\theta_t) - L(\theta^*)) \\
&\overset{(c)}{\leq} (1 - \eta\gamma)\|\theta_t - \theta^*\|_2^2 ,
\end{aligned} \tag{125}
$$

in which (a) uses Assumption 1(c), which requires that $L$ is $\gamma$-convex. (b) uses Assumption 1(a), which requires that $\nabla L$ is $G$-Lipschitz. (c) uses the condition $\eta \leq 1/G$ stated in Theorem 8. Thus

$$\|\theta_t - \eta\nabla L(\theta_t) - \theta^*\|_2 \leq \sqrt{1 - \eta\gamma}\|\theta_t - \theta^*\|_2 \leq \left(1 - \frac{1}{2}\eta\gamma\right)\|\theta_t - \theta^*\|_2 . \tag{126}$$

Therefore

$$\mathbb{E}\left[\|\theta_{t+1} - \theta^*\|_2\right] \leq \left(1 - \frac{1}{2}\eta\gamma\right)\mathbb{E}\left[\|\theta_t - \theta^*\|_2\right] + \eta D\sqrt{\frac{CT}{nm}\left(1 + \frac{d\ln n}{\epsilon^2 \wedge \epsilon}\right)}. \tag{127}$$

Repeat (127) iteratively for $t = 0, \ldots, T - 1$. Then

$$\mathbb{E}\left[\|\theta_T - \theta^*\|_2\right] \leq \left(1 - \frac{1}{2}\eta\gamma\right)^T\|\theta_0 - \theta^*\|_2 + \frac{2D}{\gamma}\sqrt{\frac{CT}{nm}\left(1 + \frac{d\ln n}{\epsilon^2 \wedge \epsilon}\right)}. \tag{128}$$

With $c_T \ln n \leq T \leq C_T \ln n$ for some constant $c_T$ and $C_T$,

$$\mathbb{E}\left[\|\theta_T - \theta^*\|_2\right] \lesssim D\sqrt{\frac{\ln n}{nm}\left(1 + \frac{d\ln n}{\epsilon^2 \wedge \epsilon}\right)}. \tag{129}$$

## E    NONPARAMETRIC CLASSIFICATION

### E.1    ALGORITHM DESCRIPTION

We state the algorithm for $\epsilon \leq 1$ first, and then extend to larger $\epsilon$.

$$K = 2^{\lceil\log_2 B\rceil} \tag{130}$$

be the minimum integer that is a power of 2 and is not smaller than $B$. Denote $\mathbf{H}_K$ as the Hadamard matrix of order $K$. Define

$$T_k = \underset{l\in[B]:H_{kl}=1}{\cup} B_l, k = 1, \ldots, K, \tag{131}$$

and

$$q_k = \begin{cases} \int_{B_k} f(\mathbf{x})\eta(\mathbf{x})d\mathbf{x} & \text{if} & k = 1, \dots, B \\ 0 & \text{if} & k = B+1, \dots, K. \end{cases} \tag{132}$$

Furthermore, define

$$Q_k = \int_{T_k} f(\mathbf{x})\eta(\mathbf{x})d\mathbf{x} - \int_{T_k^c} f(\mathbf{x})\eta(\mathbf{x})d\mathbf{x}, \tag{133}$$

in which $T_k^c$ is the complement of $T_k$. Then

$$Q_k = \sum_{l \in [B]:H_{kl}=1} q_l - \sum_{l \in [B]:H_{kl}=-1} q_l = \sum_{j=1}^{K} H_{kl}q_l. \tag{134}$$

In matrix form, we have $\mathbf{Q} = \mathbf{H}_K\mathbf{q}$, in which $\mathbf{Q} = (Q_1, \dots, Q_K)^T$, $\mathbf{q} = (q_1, \dots, q_K)^T$. Note that

$$\mathbb{E}[Y_{ij}\mathbf{1}(\mathbf{X}_{ij} \in T_k) - Y_{ij}\mathbf{1}(\mathbf{X}_{ij} \in T_k^c)] = Q_k, \tag{135}$$

thus we can just define

$$U_{ijk} = Y_{ij}\mathbf{1}(\mathbf{X}_{ij} \in T_k) - Y_{ij}\mathbf{1}(\mathbf{X}_{ij} \in T_k^c), \tag{136}$$

then we have $\mathbb{E}[U_{ijk}] = Q_k$, and $|U_{ijk}| \leq 1$. Therefore, from $U_{ijk}$, we can estimate $Q_k$ using our one dimensional mean estimation method. This approach solves the issue caused by direct extension of the algorithm in (Berrett & Butucea, 2019). Since the bound of $|U_{ijk}|$ does not increase with $m$, the strength of noise remains the same, thus the severe loss on the accuracy can be avoided.

Based on the discussions above, our detailed algorithm is described as following, and stated precisely in Algorithm 4. Right now, we focus on the case with $\epsilon \leq 1$.

**Training.** Firstly, we divide the users randomly into $K$ groups, such that the $k$-th group is used to estimate $Q_k$ using the one dimensional mean estimation method, i.e. Algorithm 1, for $k = 1, \dots, K$:

$$\hat{Q}_k = MeanEst1d(\{U_{ijk}|i \in S_k, j \in [m]\}). \tag{137}$$

$\hat{Q}_k$ with $k = 1, \dots, K$ are grouped into a vector $\hat{\mathbf{Q}} = (\hat{Q}_1, \dots, \hat{Q}_K)^T$. Then $q_k$ can be estimated using $\hat{\mathbf{Q}}$:

$$\hat{\mathbf{q}} = \mathbf{H}_K^{-1}\mathbf{Q} = \frac{1}{K}\mathbf{H}_K\hat{\mathbf{Q}}, \tag{138}$$

in which $\hat{\mathbf{q}} = (\hat{q}_1, \dots, \hat{q}_K)^T$ is the vector containing the estimate of $q_1, \dots, q_K$.

Now we comment on the privacy property of the training process. Samples are privatized in step 4, which uses Algorithm 1. According to Theorem 1, with $h = 4D/\sqrt{m}$ and $\Delta = D\sqrt{\ln n/m}$, this step satisfies user-level $\epsilon$-LDP, and thus the whole training process satisfies the privacy requirement.

**Prediction.** For any test sample $\mathbf{X}$, let the output be

$$\hat{Y} = \sum_{k=1}^{B} \text{sign}(\hat{q}_k)\mathbf{1}(\mathbf{x} \in B_k). \tag{139}$$

Finally, we extend the algorithm to larger $\epsilon$. The idea is similar to the multi-dimensional mean estimation shown in Section C.1.

*Medium privacy ($1 \leq \epsilon < K \ln n$).* The users are divided into $\lceil K/\epsilon \rceil$ groups (instead of $K$ groups for $\epsilon \leq 1$ case). The $k$-th group is used to estimate $\epsilon$ components $Q_{(k-1)\epsilon+1}, \dots, Q_{k\epsilon}$, under 1-LDP for each component.

*Low privacy ($\epsilon > K \ln n$).* In this case, do not divide users into groups. Just estimate each $Q_k$ under $\epsilon/K$-LDP.

---

**Algorithm 4** Training algorithm of nonparametric classification under user-level $\epsilon$-LDP

---

**Input:** Training dataset containing $n$ users with $m$ samples per user, i.e. $(\mathbf{X}_{ij}, Y_{ij})$, $i = 1, \ldots, n$, $j = 1, \ldots, m$
**Output:** $\hat{\mathbf{q}}$
**Parameter:** $h$, $\Delta$, $l$

1: Divide $\mathcal{X} = [0, 1]^d$ into $B$ bins, such that the length of each bin is $l$;
2: $K = 2^{\lceil \log_2 B \rceil}$;
3: Calculate $U_{ijk}$ according to (136), for $i = 1, \ldots, n$, $j = 1, \ldots, m$, $k = 1, \ldots, K$;
4: Estimate $\hat{Q}_k$ according to (137), with parameters $h$ and $\Delta$, for $k = 1, \ldots, K$;
5: $\hat{\mathbf{q}} = \mathbf{H}_K \hat{\mathbf{Q}} / K$, in which $\hat{\mathbf{Q}} = (\hat{Q}_1, \ldots, \hat{Q}_K)^T$;
6: **Return** $\hat{\mathbf{q}}$

---

### E.2 PROOF OF THEOREM 9

To begin with, we show a concentration inequality of one dimensional mean estimation.

**Lemma 6.** *Let $E_1$ be the event that stage I is successful, i.e. $\mu \in [L, R]$. For any $t \leq \sqrt{2}(3h + 2\Delta)$, in which $h = 4D/\sqrt{m}$ and $\Delta = D\sqrt{\ln(Kn)/m}$, then the following bound holds:*

$$P(|\hat{\mu} - \mu| > t | E_1) \leq 2 \exp \left[ -\frac{n \left( t - 4\sqrt{2\pi} \frac{D}{\sqrt{mnK}} \right)^2}{2 \left( \frac{1}{4} + \frac{4}{\epsilon^2} \right) (3h + 2\Delta)^2} \right]. \tag{140}$$

*Proof.* Define $a = 3h + 2\Delta$ for convenience. For $i = n/2, \ldots, n$, since $W_i \sim \text{Lap}(a/\epsilon)$,

$$\mathbb{E}[e^{\lambda W_i} | E_1] \leq \exp \left[ 2 \left( \frac{a}{\epsilon} \right)^2 \lambda^2 \right], \forall \lambda^2 \leq \frac{\epsilon^2}{2a^2}. \tag{141}$$

Similar to (33), it can be shown that

$$\mathbb{E} \left[ \exp \left[ (Y_i \vee (L - \Delta)) \wedge (L + \Delta) - \mathbb{E}[(Y_i \vee (L - \Delta)) \wedge (L + \Delta)]] \right] \right] \leq e^{\frac{1}{8} \lambda^2 a^2}. \tag{142}$$

Note that $Z_{ik} = \mathbf{1}(Y_i \in B_k) + W_{ik}$, thus for $i = n/2, \ldots, n$,

$$\mathbb{E}[e^{\lambda(Z_i - \mathbb{E}[Z_i])} | E_1] \leq \exp \left[ \left( \frac{1}{8} + \frac{2}{\epsilon^2} \right) a^2 \lambda^2 \right], \forall \lambda^2 \leq \frac{\epsilon^2}{2a^2}. \tag{143}$$

Recall that $\hat{\mu} = (2/n) \sum_{i=n/2+1}^n Z_i$,

$$\mathbb{E} \left[ e^{\lambda(\hat{\mu} - \mathbb{E}[\hat{\mu}])} | E_1 \right] \leq \exp \left[ \left( \frac{1}{8} + \frac{2}{\epsilon^2} \right) \frac{2a^2 \lambda^2}{n} \right], \forall \lambda^2 \leq \frac{n^2 \epsilon^2}{8a^2}. \tag{144}$$

Hence

$$P(\hat{\mu} - \mathbb{E}[\hat{\mu}] > t | E_1) \leq \inf_{|\lambda| \leq n\epsilon/(2\sqrt{2}a)} \exp \left[ -\lambda t + \left( \frac{1}{8} + \frac{2}{\epsilon^2} \right) \frac{2a^2 \lambda^2}{n} \right]. \tag{145}$$

If

$$t \leq \frac{\epsilon}{\sqrt{2}} \left( \frac{1}{4} + \frac{4}{\epsilon^2} \right) a, \tag{146}$$

then the right hand side of (145) reaches minimum at

$$\lambda^* = \frac{nt}{2 \left( \frac{1}{4} + \frac{4}{\epsilon^2} \right) a^2}. \tag{147}$$

The condition (146) can be simplified to $t \leq \sqrt{2}a$. It remains to consider the estimation bias. Following arguments similar to those used to derive (52), with $h = 4D/\sqrt{m}$ and $\Delta = D\sqrt{\ln(Kn)/m}$, the bias is bounded b$4\sqrt{2\pi}D/\sqrt{mnK}$. Therefore

$$P(|\hat{\mu} - \mu| > t | E_1) \leq 2 \exp \left[ -\frac{n}{2 \left( \frac{1}{4} + \frac{4}{\epsilon^2} \right) a^2} \left( t - \frac{4\sqrt{2\pi}D}{\sqrt{mnK}} \right)^2 \right]. \tag{148}$$

The proof is complete. $\qquad\square$

Now we focus on the case with $\epsilon \leq 1$. Denote $E_{1k}$ as the event that the first stage is successful for estimating $\hat{q}_k$, and $E_1 = \cap_k E_{1k}$. Recall (137) estimates $Q_k$ using Algorithm 1. From Lemma 6, the following lemma can be proved easily:

**Lemma 7.** *There exists two constants $C_1$, $C_2$, such that*

$$P(|\hat{q}_k - q_k| > t | E_1) \leq 2 \exp\left[-C_1 \frac{mn\epsilon^2}{\ln(nK)}\left(t - C_2\sqrt{\frac{1}{mn}}\right)^2\right]. \tag{149}$$

*Proof.* The size of the $k$-th group is $|S_k| = n/K$, from Lemma 6, since $U_{ijk}$ in (136) satisfies $|U_{ijk}| \leq 1$, the following bound holds:

$$P(|\hat{Q}_k - Q_k| > t | E_1) \leq 2 \exp\left[-\frac{n\left(t - 4\sqrt{2\pi}\sqrt{\frac{1}{mn}}\right)^2}{2K\left(\frac{1}{4} + \frac{4}{\epsilon^2}\right)(3h + 2\Delta)^2}\right]. \tag{150}$$

From (138),

$$|\hat{q}_k - q_k| = \left|\frac{1}{K}\sum_{l=1}^{K}\mathbf{H}_{kl}(\hat{Q}_l - Q_l)\right| \tag{151}$$

Note that $\hat{Q}_l$ are independent for different $l$, and the values of $\mathbf{H}_{kl}$ are either $1$ or $-1$. Moreover, as discussed in Section 4, $h \sim 1/\sqrt{m}$, $\Delta \sim \sqrt{\ln n/m}$, there exists a constant $C_1$ and $C_2$ such that (149) holds. $\square$

From (42), the failure probability of the first stage is bounded by

$$P(E_{1k}^c) \leq \sqrt{m}e^{-c_0 n\epsilon^2}. \tag{152}$$

We then bound the excess risk of classification. Suppose $\mathbf{x} \in B_k$. Then given $\mathbf{x}$ and $\hat{q}_k$ obtained from training samples,

$$
\begin{aligned}
P(\hat{Y} \neq Y | \mathbf{x}, \hat{q}_k) &= P(Y \neq \text{sign}(\hat{q}_k)) \\
&\leq \mathbf{1}(\text{sign}(\hat{q}_k) \neq \text{sign}(\eta(\mathbf{x})))P(Y = \text{sign}(\eta(\mathbf{x}))) \\
&\quad + \mathbf{1}(\text{sign}(\hat{q}_k) = \text{sign}(\eta(\mathbf{x})))P(Y \neq \text{sign}(\eta(\mathbf{x}))) \\
&= \mathbf{1}(\text{sign}(\hat{q}_k) \neq \text{sign}(\eta(\mathbf{x})))\frac{|\eta(\mathbf{x})| + 1}{2} + \mathbf{1}(\text{sign}(\hat{q}_k) = \text{sign}(\eta(\mathbf{x})))\frac{1 - |\eta(\mathbf{x})|}{2} \\
&= \frac{1 - |\eta(\mathbf{x})|}{2} + |\eta(\mathbf{x})|\mathbf{1}(\text{sign}(\hat{q}_k) \neq \text{sign}(\eta(\mathbf{x}))). 
\end{aligned}
\tag{153}
$$

Therefore

$$R = \mathbb{E}\left[\frac{1 - |\eta(\mathbf{X})|}{2}\right] + \mathbb{E}\left[\sum_{k=1}^{B}\int_{B_k}|\eta(\mathbf{x})|\mathbf{1}(\text{sign}(\hat{q}_k) \neq \text{sign}(\eta(\mathbf{x})))f(\mathbf{x})d\mathbf{x}\right]. \tag{154}$$

Recall that the Bayes risk is

$$R^* = \mathbb{E}\left[\frac{1 - |\eta(\mathbf{X})|}{2}\right], \tag{155}$$

thus the excess risk is

$$R - R^* = \mathbb{E}\left[\sum_{k=1}^{B}\int_{B_k}|\eta(\mathbf{x})|\mathbf{1}(\text{sign}(\hat{q}_k) \neq \text{sign}(\eta(\mathbf{x})))f(\mathbf{x})d\mathbf{x}\right]. \tag{156}$$

Define

$$\eta_0 = 2C_b d^{\frac{\beta}{2}}l^\beta + \frac{2C_2}{f_L l^d}\sqrt{\frac{1}{mn}}. \tag{157}$$

If $\eta(\mathbf{x}) > \eta_0$, then

$$q_k = \int_{B_k} f(\mathbf{x})\eta(\mathbf{x})d\mathbf{x} \geq \left(\int_{B_k} f(\mathbf{x})d\mathbf{x}\right)\left(\eta(\mathbf{x}) - C_b d^{\frac{\beta}{2}} l^\beta\right) > 0. \tag{158}$$

Similarly, if $\eta(\mathbf{x}) < -\eta_0$, $q_k < 0$. Thus $\mathrm{sign}(\eta(\mathbf{x})) = \mathrm{sign}(q_k)$ if $|\eta(\mathbf{x})| > \eta_0$. Therefore, for all $\mathbf{x}$ such that $\eta(\mathbf{x}) > \eta_0$,

$$\begin{aligned}
\mathrm{P}(\mathrm{sign}(\hat{q}_k) \neq \mathrm{sign}(\eta(\mathbf{x}))) &\leq \mathrm{P}(\mathrm{sign}(\hat{q}_k) \neq \mathrm{sign}(q_k)) \\
&\leq \mathrm{P}(E_1^c) + \mathrm{P}(|\hat{q}_k - q_k| > |q_k||E_1) \\
&\overset{(a)}{\leq} \sqrt{m}e^{-c_0 n\epsilon^2} + 2\exp\left[-C_1 \frac{mn\epsilon^2}{\ln n}\left(|q_k| - \frac{C_2}{\sqrt{mn}}\right)^2\right] \\
&\overset{(b)}{\leq} \sqrt{m}e^{-c_0 n\epsilon^2} + 2\exp\left[-\frac{1}{4}C_1 \frac{mn\epsilon^2}{\ln n}|q_k|^2\right] \\
&\overset{(c)}{\leq} \sqrt{m}e^{-c_0 n\epsilon^2} + 2\exp\left[-\frac{1}{16}C_1 f_L^2 \frac{mn\epsilon^2}{\ln n}\eta^2(\mathbf{x})l^{2d}\right]. \tag{159}
\end{aligned}$$

Now we explain (a)-(c) in (159). (a) uses (152) and Lemma 7. For (b), note that with $\eta(\mathbf{x}) > \eta_0$,

$$\begin{aligned}
|q_k| &= \left|\int_{B_k} \eta(\mathbf{x})f(\mathbf{x})d\mathbf{x}\right| \\
&\geq |\eta(\mathbf{x}) - C_b d^{\frac{\beta}{2}} l^\beta| \int_{B_k} f(\mathbf{x})d\mathbf{x} \\
&\geq (\eta_0 - C_b d^{\frac{\beta}{2}} l^\beta) \int_{B_k} f(\mathbf{x})d\mathbf{x} \\
&\geq \frac{2C_2}{f_L l^d}\sqrt{\frac{1}{mn}} \int_{B_k} f(\mathbf{x})d\mathbf{x} \\
&\geq 2C_2\sqrt{\frac{1}{mn}}. \tag{160}
\end{aligned}$$

Thus

$$|q_k| - \frac{C_2}{\sqrt{mn}} \geq \frac{1}{2}|q_k|. \tag{161}$$

For (c), since $|\eta(\mathbf{x})| > \eta_0 > 2C_b d^{\frac{\beta}{2}} l^\beta$,

$$\eta(\mathbf{x}) - C_b d^{\frac{\beta}{2}} l^\beta > \frac{1}{2}\eta(\mathbf{x}). \tag{162}$$

Hence

$$|q_k| \geq \frac{1}{2}\eta(\mathbf{x}) \int_{B_k} f(\mathbf{x})d\mathbf{x} \geq \frac{1}{2}\eta(\mathbf{x})f_L l^d. \tag{163}$$

The proof of (159) (a)-(c) are complete. For $\mathbf{x} \in B_k$, denote $\hat{\eta}(\mathbf{x}) = q_k$. Then based on (159),

$$\begin{aligned}
R - R^* &= \sum_{k=1}^{B} \int_{B_k} |\eta(\mathbf{x})|\mathrm{P}(\mathrm{sign}(\hat{q}_k) \neq \mathrm{sign}(\eta(\mathbf{x})))f(\mathbf{x})d\mathbf{x} \\
&= \int_{\eta(\mathbf{x}) \leq \eta_0} \eta_0 f(\mathbf{x})d\mathbf{x} + \int_{\eta(\mathbf{x}) > \eta_0} |\eta(\mathbf{x})|\mathrm{P}(\mathrm{sign}(\hat{\eta}(\mathbf{x})) \neq \mathrm{sign}(\eta(\mathbf{x})))f(\mathbf{x})d\mathbf{x} \\
&\leq \eta_0 \mathrm{P}(\eta(\mathbf{X}) < \eta_0) + 2\mathbb{E}\left[|\eta(\mathbf{X})|\exp\left[-\frac{1}{16}C_1 f_L^2 \frac{mn\epsilon^2}{\ln n}\eta^2(\mathbf{x})l^{2d}\right]\right] + \sqrt{m}e^{-c_0 n\epsilon^2}. \tag{164}
\end{aligned}$$

For the first term in (164), use Assumption 2, we have

$$\mathrm{P}(\eta(\mathbf{X}) < \eta_0) \lesssim \eta_0^\gamma. \tag{165}$$

For the second term, we can bound it with Lemma 11. The third term decays exponentially with $n$. Therefore, with $n\epsilon^2 \gtrsim \ln m$, we have

$$
\begin{aligned}
R - R^* &\lesssim \eta_0^{1+\gamma} + \left(\frac{mn\epsilon^2}{\ln n} l^{2d}\right)^{-\frac{1}{2}(1+\gamma)} \\
&\sim \left(l^\beta + \frac{1}{l^d}\sqrt{\frac{1}{mn}} + \frac{\ln n}{\sqrt{mn\epsilon^2 l^d}}\right)^{1+\gamma},
\end{aligned} \tag{166}
$$

in which the second step uses (157). Let

$$
l \sim (mn\epsilon^2)^{-\frac{1}{2(d+\beta)}}, \tag{167}
$$

then

$$
R - R^* \lesssim (mn\epsilon^2)^{-\frac{\beta(1+\gamma)}{2(d+\beta)}} \ln^{1+\gamma} n. \tag{168}
$$

Now the proof of the bound of mean squared error for $\epsilon \leq 1$ is finished. It remains to show the case with $\epsilon > 1$.

*1) Medium privacy ($1 \leq \epsilon < K \ln n$).* Note that now the size of each group is $n/\lceil K/\epsilon \rceil$. Following the arguments above, it can be shown that with $l \sim (mn\epsilon^2)^{-\frac{1}{2(d+\beta)}}$,

$$
R - R^* \lesssim (mn\epsilon)^{-\frac{\beta(1+\gamma)}{2(d+\beta)}} \ln^{1+\gamma} n. \tag{169}
$$

*2) Low privacy ($\epsilon \geq K \ln n$).* Now (149) becomes

$$
\mathrm{P}(|\hat{q}_k - q_k| > t | E_1) \leq 2\exp\left[-C_1 \frac{mnK}{\ln n}\left(t - C_2\sqrt{\frac{1}{mnK}}\right)^2\right]. \tag{170}
$$

Following previous arguments,

$$
R - R^* \lesssim \left(l^\beta + \frac{1}{l^d}\sqrt{\frac{\ln n}{nmK}}\right)^{1+\gamma}. \tag{171}
$$

With $l \sim (nm/\ln n)^{-1/(2\beta+d)}$,

$$
R - R^* \lesssim \left(\frac{nm}{\ln n}\right)^{-\frac{\beta(1+\gamma)}{2\beta+d}}. \tag{172}
$$

Combine (168), (169) and (172), the final bound on mean squared error is

$$
R - R^* \lesssim (mn(\epsilon^2 \wedge \epsilon))^{-\frac{\beta(1+\gamma)}{2(d+\beta)}} \ln^{1+\gamma} n + \left(\frac{nm}{\ln n}\right)^{-\frac{\beta(1+\gamma)}{2\beta+d}}. \tag{173}
$$

### E.3 PROOF OF THEOREM 10

Divide the whole support into $B$ bins, and the length of each bin is $l$. Then $Bl^d = 1$. Let the pdf of $\mathbf{X}$ be uniform, i.e. $f(\mathbf{x}) = c$ for some constant $c$. Moreover, let $\phi(\mathbf{u})$ be some function supported at $[-1/2, 1/2]^d$, such that $\phi(\mathbf{u}) \geq 0$ and $\phi(\mathbf{u})l^\beta \leq 1/2$ always hold, and for any $\mathbf{x}$ and $\mathbf{x}'$,

$$
\|\phi(\mathbf{u}) - \phi(\mathbf{u}')\| \leq C_b \|\mathbf{u} - \mathbf{u}'\|^\beta. \tag{174}
$$

Moreover, denote $\mathbf{c}_1, \ldots, \mathbf{c}_K$ be centers of $K$ bins, $K < B$. For $\mathbf{v} \in \mathcal{V} := \{-1, 1\}^K$, let

$$
\eta_\mathbf{v}(\mathbf{x}) = \sum_{k=1}^K v_k \phi\left(\frac{\mathbf{x} - \mathbf{c}_k}{l}\right) l^\beta. \tag{175}
$$

For other $B - K$ bins, $\eta(\mathbf{x}) = 0$. It can be proved that there exists a constant $C_K$, such that if $K \leq C_K l^{\gamma\beta-d}$, then $\eta(\mathbf{x})$ satisfies Assumption 2.

Denote

$$\hat{v}_k = \arg\max_{s \in \{-1,1\}} \int_{B_k} \phi\left(\frac{\mathbf{x} - \mathbf{c}_k}{l}\right) \mathbf{1}(\text{sign}(\hat{\eta}(\mathbf{x})) = s)f(\mathbf{x})d\mathbf{x}. \tag{176}$$

If $\hat{v}_k \neq v_k$, then

$$\int_{B_k} \phi\left(\frac{\mathbf{x} - \mathbf{c}_k}{l}\right) \mathbf{1}(\text{sign}(\hat{\eta}(\mathbf{x})) = v_k)f(\mathbf{x})d\mathbf{x} \leq \int_{B_k} \phi\left(\frac{\mathbf{x} - \mathbf{c}_k}{l}\right) \mathbf{1}(\text{sign}(\hat{\eta}(\mathbf{x})) = -v_k)f(\mathbf{x})d\mathbf{x}. \tag{177}$$

Note that

$$\int_{B_k} \phi\left(\frac{\mathbf{x} - \mathbf{c}_k}{l}\right) \left[\mathbf{1}(\text{sign}(\hat{\eta}(\mathbf{x})) = v_k) + \mathbf{1}(\text{sign}(\hat{\eta}(\mathbf{x})) = -v_k)\right] f(\mathbf{x})d\mathbf{x} = \int_{B_k} \phi\left(\frac{\mathbf{x} - \mathbf{c}_k}{l}\right) f(\mathbf{x})d\mathbf{x}$$

$$\geq cl^d \int \phi(\mathbf{u})d\mathbf{u} = cl^d \|\phi\|_1. \tag{178}$$

Therefore, if $\hat{v}_k \neq v_k$, then from (177) and (178),

$$\int_{B_k} \phi\left(\frac{\mathbf{x} - \mathbf{c}_k}{l}\right) \mathbf{1}(\text{sign}(\hat{\eta}(\mathbf{x})) = -v_k)f(\mathbf{x})d\mathbf{x} \geq \frac{1}{2}cl^d \|\phi\|_1. \tag{179}$$

Denote the vector form $\hat{\mathbf{v}} = (\hat{v}_1, \ldots, \hat{v}_k)$. Then the Bayes risk is bounded by

$$
\begin{aligned}
R - R^* &= \int |\eta_{\mathbf{v}}(\mathbf{x})| \mathbf{P}(\text{sign}(\hat{\eta}(\mathbf{x})) \neq \text{sign}(\eta_{\mathbf{v}}(\mathbf{x})))f(\mathbf{x})d\mathbf{x} \\
&= \sum_{k=1}^{K} \int_{B_k} |\eta_{\mathbf{v}}(\mathbf{x})| \mathbf{P}(\text{sign}(\hat{\eta}(\mathbf{x})) = -v_k)f(\mathbf{x})d\mathbf{x} \\
&= l^\beta \sum_{k=1}^{K} \mathbb{E}\left[\int_{B_k} \phi\left(\frac{\mathbf{x} - \mathbf{c}_k}{l}\right) \mathbf{1}(\text{sign}(\hat{\eta}(\mathbf{x}) = -v_k))f(\mathbf{x})d\mathbf{x}\right] \\
&\geq \frac{1}{2}cl^{\beta+d} \|\phi\|_1 \mathbb{E}[\rho_H(\hat{\mathbf{v}}, \mathbf{v})],
\end{aligned}
\tag{180}
$$

in which $\rho_H(\hat{\mathbf{v}}, \mathbf{v})$ is the Hamming distance. Hence

$$\inf_{\hat{Y}} \inf_{Q \in \mathcal{Q}_\epsilon} \sup_{(p,\eta) \in \mathcal{P}_{cls}} (R - R^*) \geq \frac{1}{2}cl^{\beta+d} \|\phi\|_1 \inf_{\hat{\mathbf{v}}} \inf_{Q \in \mathcal{Q}_\epsilon} \sup_{\mathbf{v} \in \mathcal{V}} \mathbb{E}[\rho_H(\hat{\mathbf{v}}, \mathbf{v})]. \tag{181}$$

Define

$$\delta = \sup_{Q \in \mathcal{Q}_\epsilon} \max_{\mathbf{v}, \mathbf{v}': \rho_H(\mathbf{v}, \mathbf{v}') = 1} D(p_{\mathbf{Z}|\mathbf{v}} || p_{\mathbf{Z}|\mathbf{v}'}), \tag{182}$$

in which $p_{\mathbf{Z}|\mathbf{v}}$ denotes the distribution of privatized variable $\mathbf{Z}$ given $\eta = \eta_{\mathbf{v}}$. From (Tsybakov, 2009), Theorem 2.12(iv),

$$\inf_{\hat{\mathbf{v}}} \sup_{\mathbf{v} \in \mathcal{V}} \mathbb{E}[\rho_H(\hat{\mathbf{v}}, \mathbf{v})] \geq \frac{K}{2} \max\left(\frac{1}{2}e^{-\delta}, 1 - \sqrt{\frac{\delta}{2}}\right). \tag{183}$$

It remains to bound $\delta$, i.e. $D(p_{\mathbf{Z}|\mathbf{v}} || p_{\mathbf{Z}|\mathbf{v}'})$ under the constraint that $\rho_H(\mathbf{v}, \mathbf{v}') = 1$. From (Duchi et al., 2018), Theorem 1, we have

$$D(p_{\mathbf{Z}|\mathbf{v}} || p_{\mathbf{Z}|\mathbf{v}'}) \leq n(e^\epsilon - 1)^2 \mathbb{TV}^2(p_{\mathbf{v}}^m, p_{\mathbf{v}'}^m), \tag{184}$$

in which $p_{\mathbf{v}}^m$ denotes the joint distribution of $(\mathbf{X}, Y)$ (i.e. before privatization) given $\eta = \eta_{\mathbf{v}}$. Note that $p_{\mathbf{v}}$ and $p_{\mathbf{v}'}$ are only different in one bin. Without loss of generality, suppose that $p_{\mathbf{v}}$ and $\mathbf{v}'$ are

different at the first bin. Using Lemma 10, we have

$$\mathbb{TV}^2(p_{\mathbf{v}}^m, p_{\mathbf{v}'}^m)$$

$$\overset{(a)}{\leq} \frac{1}{2} p_{\mathbf{v}}(\mathbf{X} \in B_1) D(p_{\mathbf{v}}^m \| p_{\mathbf{v}'}^m)$$

$$= \frac{1}{2} l^d D(p_{\mathbf{v}}^m \| p_{\mathbf{v}'}^m)$$

$$\leq \frac{1}{2} m l^d D(p_{\mathbf{v}} \| p_{\mathbf{v}'})$$

$$= \frac{1}{2} m l^d \int_{B_1} f(\mathbf{x}) \left[ p_{\mathbf{v}}(Y = 1|\mathbf{x}) \ln \frac{p_{\mathbf{v}}(Y = 1|\mathbf{x})}{p_{\mathbf{v}'}(Y = 1|\mathbf{x})} + p_{\mathbf{v}}(Y = -1|\mathbf{x}) \ln \frac{p_{\mathbf{v}}(Y = -1|\mathbf{x})}{p_{\mathbf{v}'}(Y = -1|\mathbf{x})} \right] d\mathbf{x}$$

$$\overset{(b)}{=} \frac{1}{2} m l^d \int_{B_1} f(\mathbf{x}) \left[ \frac{1 + \eta_{\mathbf{v}}(\mathbf{x})}{2} \ln \frac{1 + \eta_{\mathbf{v}}(\mathbf{x})}{1 - \eta_{\mathbf{v}}(\mathbf{x})} + \frac{1 - \eta_{\mathbf{v}}(\mathbf{x})}{2} \ln \frac{1 - \eta_{\mathbf{v}}(\mathbf{x})}{1 + \eta_{\mathbf{v}}(\mathbf{x})} \right] d\mathbf{x}$$

$$= \frac{1}{2} m l^d \int_{B_1} f(\mathbf{x}) \eta_{\mathbf{v}}(\mathbf{x}) \ln \frac{1 + \eta_{\mathbf{v}}(\mathbf{x})}{1 - \eta_{\mathbf{v}}(\mathbf{x})} d\mathbf{x}$$

$$\overset{(c)}{\leq} \frac{3}{2} m l^d \int_{B_1} f(\mathbf{x}) \eta_{\mathbf{v}}^2(\mathbf{x}) d\mathbf{x}$$

$$\leq \frac{3}{2} m l^{d+2\beta} \int_{B_1} \phi^2 \left( \frac{\mathbf{x} - \mathbf{c}_j}{h} \right) d\mathbf{x}$$

$$= \frac{3}{2} m l^{2d+2\beta} \| \phi \|_2^2. \tag{185}$$

(a) holds because $p_{\mathbf{v}}$ and $p_{\mathbf{v}'}$ are only different at $B_1$. For (b), recall that $\eta(\mathbf{x}) = \mathbb{E}[Y|\mathbf{X} = \mathbf{x}]$. (c) holds since $|\eta_{\mathbf{v}}(\mathbf{x})| \leq 1/2$ (recall the condition $\phi(\mathbf{u}) l^\beta \leq 1/2$), if $v_1 = 1$, then $\ln(1 + \eta_{\mathbf{v}}(\mathbf{x})) \leq \eta_{\mathbf{v}}(\mathbf{x})$, $\ln(1/(1 - \eta_{\mathbf{v}}(\mathbf{x}))) \leq 2\eta_{\mathbf{v}}(\mathbf{x})$. Similar result can be obtained for $v_1 = -1$. From (182) and (184),

$$\delta \leq \frac{3}{2} n(e^\epsilon - 1)^2 m l^{2d+2\beta} \| \phi \|_2^2. \tag{186}$$

Let

$$l \sim (nm\epsilon^2)^{-\frac{1}{2(d+\beta)}}, \tag{187}$$

then $\delta \lesssim 1$. Moreover, let $K \sim l^{\gamma\beta - d}$, then

$$\inf_{\hat{\mathbf{v}}} \sup_{\mathbf{v} \in \mathcal{V}} \mathbb{E}[\rho_H(\hat{\mathbf{v}}, \mathbf{v})] \gtrsim K \sim l^{\gamma\beta - d}. \tag{188}$$

From (181),

$$\inf_{\hat{Y}} \inf_{Q \in \mathcal{Q}_\epsilon} \sup_{(p,\eta) \in \mathcal{P}_{cls}} (R - R^*) \gtrsim l^{\beta+d} l^{\gamma\beta - d} = l^{\beta(1+\gamma)} \sim (nm\epsilon^2)^{-\frac{\beta(1+\gamma)}{2(d+\beta)}}. \tag{189}$$

# F    NONPARAMETRIC REGRESSION

## F.1    ALGORITHM DESCRIPTION

Define $q_k$ and $Q_k$ in the same way as (132) and (133). Moreover, define $p_k = \int_{B_k} f(\mathbf{x}) d\mathbf{x}$, and

$$P_k = \int_{T_k} f(\mathbf{x}) d\mathbf{x} - \int_{T_k^c} f(\mathbf{x}) d\mathbf{x}, \tag{190}$$

in which $T_k$ is defined in (131), and $T_k^c$ is the complement.

Denote

$$\eta_k := \frac{q_k}{p_k} = \frac{\int_{B_k} f(\mathbf{x}) \eta(\mathbf{x}) d\mathbf{x}}{\int_{B_k} f(\mathbf{x}) d\mathbf{x}}, \tag{191}$$

then $\eta_k$ can be viewed as the average of $\eta(\mathbf{x})$ weighted by the pdf. If $\eta$ is continuous and $l$ is sufficiently small, then $\eta(\mathbf{x}) \approx \eta_k$ for all $\mathbf{x} \in B_k$. Hence, for any $\mathbf{x} \in B_k$, we can just estimate $\eta(\mathbf{x})$ by estimating $q_k$ and $p_k$. As has been discussed in the classification case, direct estimation is not efficient. Therefore, we estimate $\mathbf{Q} = (Q_1, \ldots, Q_K)$ and $\mathbf{P} = (P_1, \ldots, P_k)$ first, and then calculate $q_k$ and $p_k$ for $k = 1, \ldots, K$.

**Training.** Recall that in the classification problem, we have divided the dataset into $K$ parts, which are used to estimate $Q_k$ for $k = 1, \ldots, K$ respectively. For regression problem, we need to estimate both $Q_k$ and $P_k$. Therefore, now we divide the samples randomly into $2K$ groups, such that $K$ groups are used to estimate $Q_k$, $k = 1, \ldots, K$, while the other $K$ groups are used to estimate $P_k$. The detailed steps are similar to the classification problem. In particular, $U_{ijk}$ is still calculated using (136). Since $\mathbb{E}[U_{ijk}] = Q_k$, $Q_k$ can still be estimated using (137). To estimate $P_k$, let

$$V_{ijk} = \mathbf{1}(\mathbf{X}_{ij} \in T_k) - \mathbf{1}(\mathbf{X}_{ij} \in T_k^c). \tag{192}$$

Then we have $\mathbb{E}[V_{ijk}] = P_k$, and $|V_{ijk}| \leq 1$. Therefore, $P_k$ can be estimated similarly for $k = 1, \ldots, K$:

$$\hat{P}_k = MeanEst1d(\{V_{ijk} | i \in S_{K+k}, j \in [m]\}). \tag{193}$$

Note that samples are privatized in this step. With appropriate parameters, our method satisfies user-level $\epsilon$-LDP. Based on the values of $\hat{Q}_k$ and $\hat{P}_k$ for $k = 1, \ldots, K$, $q_k$ and $p_k$ can be estimated by

$$\hat{\mathbf{q}} = \frac{1}{K}\mathbf{H}\hat{\mathbf{Q}}, \hat{\mathbf{p}} = \frac{1}{K}\mathbf{H}\hat{\mathbf{P}}, \tag{194}$$

in which $\hat{\mathbf{q}} = (\hat{q}_1, \ldots, \hat{q}_K)$, $\hat{\mathbf{p}} = (\hat{p}_1, \ldots, \hat{p}_K)$.

**Prediction.** For any test sample at $\mathbf{x} \in B_k$, The regression output is

$$\hat{\eta}(\mathbf{x}) = \frac{\hat{q}_k}{\hat{p}_k}. \tag{195}$$

The whole training algorithm is summarized in Algorithm 5.

---

**Algorithm 5** Training algorithm of nonparametric regression under user-level $\epsilon$-LDP

---

**Input:** Training dataset containing $n$ users with $m$ samples per user, i.e. $(\mathbf{X}_{ij}, Y_{ij})$, $i = 1, \ldots, n$, $j = 1, \ldots, m$
**Output:** $\hat{\mathbf{q}}$, $\hat{\mathbf{p}}$
**Parameter:** $h_q$, $h_p$, $\Delta_q$, $\Delta_p$, $l$
 Divide $\mathcal{X} = [0,1]^d$ into $B$ bins, such that the length of each bin is $l$;
 $K = 2^{\lceil \log_2 B \rceil}$;
 Calculate $U_{ijk}$ according to (136), for $i = 1, \ldots, n$, $j = 1, \ldots, m$, $k = 1, \ldots, K$;
 Calculate $V_{ijk}$ according to (192), for $i = 1, \ldots, n$, $j = 1, \ldots, m$, $k = 1, \ldots, K$;
 Estimate $\hat{Q}_k$ using (137) with parameters $h_q$ and $\Delta_q$, for $k = 1, \ldots, K$;
 Estimate $\hat{P}_k$ using (193) with parameters $h_p$ and $\Delta_p$, for $k = 1, \ldots, K$;
 $\hat{\mathbf{q}} = \mathbf{H}_K \hat{\mathbf{Q}}/K$, in which $\hat{\mathbf{Q}} = (\hat{Q}_1, \ldots, \hat{Q}_K)^T$;
 $\hat{\mathbf{p}} = \mathbf{H}_K \hat{\mathbf{P}}/K$, in which $\hat{\mathbf{P}} = (\hat{P}_1, \ldots, \hat{P}_K)^T$;
 **Return** $\hat{\mathbf{q}}$, $\hat{\mathbf{p}}$

---

### F.2 PROOF OF THEOREM 11

Define

$$\eta_k := \frac{q_k}{p_k} \tag{196}$$

Recall the definition of $q_k$ and $p_k$, we have

$$\eta_k = \frac{\int_{B_k} f(\mathbf{x})\eta(\mathbf{x})d\mathbf{x}}{\int_{B_k} f(\mathbf{x})d\mathbf{x}}, \tag{197}$$

and

$$\tilde{\eta}(\mathbf{x}) = \sum_{k=1}^{K} \eta_k \mathbf{1}(\mathbf{x} \in B_k). \tag{198}$$

Then

$$\begin{aligned} R &= \int (\hat{\eta}(\mathbf{x}) - \eta(\mathbf{x}))^2 f(\mathbf{x}) d\mathbf{x} \\ &\leq 2\mathbb{E}\left[\int (\hat{\eta}(\mathbf{x}) - \tilde{\eta}(\mathbf{x}))^2 f(\mathbf{x}) d\mathbf{x} + 2\int (\tilde{\eta}(\mathbf{x}) - \eta(\mathbf{x}))^2 f(\mathbf{x}) d\mathbf{x}\right]. \end{aligned} \tag{199}$$

The second term can be bounded with the following lemma.

**Lemma 8.**

$$\int (\tilde{\eta}(\mathbf{x}) - \eta(\mathbf{x}))^2 f(\mathbf{x}) d\mathbf{x} \leq C_b^2 d^\beta l^{2\beta}. \tag{200}$$

*Proof.*

$$\begin{aligned} \int (\tilde{\eta}(\mathbf{x}) - \eta(\mathbf{x}))^2 f(\mathbf{x}) d\mathbf{x} &= \sum_{k=1}^{K} \int_{B_k} (\eta_k - \eta(\mathbf{x}))^2 f(\mathbf{x}) d\mathbf{x} \\ &\leq \sum_{k=1}^{K} \int_{B_k} \left(C_b(\sqrt{d}l)^\beta\right)^2 f(\mathbf{x}) d\mathbf{x} \\ &= C_b^2 d^\beta l^{2\beta} \sum_{k=1}^{K} p_k \\ &= C_b d^\beta l^{2\beta}. \end{aligned} \tag{201}$$

$\square$

It remains to bound the first term of (199).

**Lemma 9.** *Denote $E_{1qk}$ and $E_{1pk}$ as the event that the first stage in estimating $Q_k$ and $P_K$ are successful, respectively. Denote $E_{1q} = \cap_k E_{1qk}$, $E_{1p} = \cap_k E_{1pk}$. Then there exists two constants $C_1$ and $C_2$, such that*

$$P(|\hat{q}_k - q_k| > t|E_{1q}) \leq 2\exp\left[-C_1 \frac{mn\epsilon^2}{T^2 \ln n}\left(t - C_2\sqrt{\frac{T}{mn}}\right)^2\right], \tag{202}$$

*and*

$$P(|\hat{p}_k - p_k| > t|E_{1p}) \leq 2\exp\left[-C_1 \frac{mn\epsilon^2}{\ln n}\left(t - C_2\sqrt{\frac{1}{mn}}\right)^2\right]. \tag{203}$$

Denote

$$\hat{\eta}_k = \frac{\hat{q}_k}{\hat{p}_k}. \tag{204}$$

Pick some constant $c$ such that $C_1 c^2 > 1$, then define

$$\begin{aligned} t_p &= C_2\sqrt{\frac{1}{mn}} + \frac{c\ln n}{\sqrt{mn\epsilon^2}}, \tag{205} \\ t_q &= C_2\sqrt{\frac{T}{mn}} + \frac{cT\ln n}{\sqrt{mn\epsilon^2}}. \tag{206} \end{aligned}$$

Then

$$P(|\hat{p}_k - p_k| > t_p|E_{1p}) \leq 2e^{-C_1 c^2 \ln n} = 2n^{-C_1 c^2}, \tag{207}$$

and

$$P(|\hat{q}_k - q_k| > t_q|E_{1q}) \leq 2n^{-C_1 c^2}. \tag{208}$$

Denote $E$ as the event that for all $k$, $|\hat{p}_k - p_k| > t_p$, $\hat{q}_k - q_k > t_q$. Then

$$
\begin{aligned}
P(E^c) &= P(\exists k, |\hat{p}_k - p_k| > t_p \text{ or } |\hat{q}_k - q_k| > t_q) \\
&\leq 4Bn^{-C_1 c^2} + P\left(E_{p1}^c \cup E_{q1}^c\right) \\
&\leq 4B\left(n^{-C_1 c^2} + \sqrt{m}e^{-C_0 n\epsilon^2}\right).
\end{aligned}
\tag{209}
$$

Hence

$$\mathbb{E}\left[\int (\hat{\eta}(\mathbf{x}) - \tilde{\eta}(\mathbf{x}))^2 f(\mathbf{x})d\mathbf{x}\mathbf{1}(E^c)\right] \leq T^2 P(E^c) \leq 4BT^2\left(n^{-C_1 c^2} + \sqrt{m}e^{-C_0 n\epsilon^2}\right). \tag{210}$$

With $C_1 c^2 \geq 1$, this term does not dominate.

Under $E$, we have

$$
\begin{aligned}
\int (\hat{\eta}(\mathbf{x}) - \tilde{\eta}(\mathbf{x}))^2 f(\mathbf{x})d\mathbf{x} &= \sum_{k=1}^{K} \int_{B_k} (\hat{\eta}(\mathbf{x}) - \eta_k)^2 f(\mathbf{x})d\mathbf{x} \\
&= \sum_{k=1}^{K} (\hat{\eta}_k - \eta_k)^2 \int_{B_k} f(\mathbf{x})d\mathbf{x} \\
&= \sum_{k=1}^{K} p_k (\hat{\eta}_k - \eta_k)^2.
\end{aligned}
\tag{211}
$$

$\hat{\eta}_k - \eta_k$ can be bounded in both two sides:

$$
\begin{aligned}
\hat{\eta}_k - \eta_k &= \frac{\hat{q}_k}{\hat{p}_k} \wedge T - \frac{q_k}{p_k} \\
&\leq \frac{q_k + t_q}{p_k - t_p} - \frac{q_k}{p_k} = \frac{p_k t_q + q_k t_p}{p_k(p_k - t_p)},
\end{aligned}
\tag{212}
$$

and

$$
\begin{aligned}
\hat{\eta}_k - \eta_k &\geq \frac{q_k - t_q}{p_k + t_p} - \frac{q_k}{p_k} \\
&= -\frac{p_k t_q + q_k t_p}{p_k(p_k + t_p)}.
\end{aligned}
\tag{213}
$$

Note that $f(\mathbf{x}) \geq f_L$, thus $p_k \geq f_L l^d$. Ensure that $l$ is picked such that $f_L l^d \geq 2t_p$. Then

$$|\hat{\eta}_k - \eta_k| \leq 2\frac{p_k t_q + q_k t_p}{p_k^2}, \tag{214}$$

$$(\hat{\eta}_k - \eta_k)^2 \leq 8\left(\frac{t_q^2}{p_k^2} + \frac{q_k^2 t_p^2}{p_k^4}\right). \tag{215}$$

Hence

$$
\begin{aligned}
\mathbb{E}\left[\int (\hat{\eta}(\mathbf{x}) - \tilde{\eta}(\mathbf{x}))^2 f(\mathbf{x})d\mathbf{x}\mathbf{1}(E)\right] &\leq \sum_{k=1}^{K} p_k \left(\frac{t_q^2}{p_k^2} + \frac{q_k^2 t_p^2}{p_k^4}\right) \\
&\lesssim \frac{\ln^2 n}{mn\epsilon^2 l^{2d}}.
\end{aligned}
\tag{216}
$$

From (210) and (216),

$$\mathbb{E}\left[(\hat{\eta}(\mathbf{x}) - \tilde{\eta}(\mathbf{x}))^2 f(\mathbf{x}) d\mathbf{x}\right] \lesssim \frac{\ln^2 n}{mn\epsilon^2 l^{2d}}. \tag{217}$$

From (199), (201) and (217),

$$R \lesssim \frac{\ln^2 n}{mn\epsilon^2 l^{2d}} + l^{2\beta}. \tag{218}$$

Let

$$l \sim \left(\frac{mn\epsilon^2}{\ln^2 n}\right)^{-\frac{1}{2(d+\beta)}}, \tag{219}$$

then

$$R \lesssim \left(\frac{mn\epsilon^2}{\ln^2 n}\right)^{-\frac{\beta}{d+\beta}}. \tag{220}$$

### F.3 PROOF OF THEOREM 12

Similar to the classification case, divide support $\mathcal{X} = [0,1]^d$ into $B$ bins with length $l$, then $Bl^d = 1$. Let $\phi(\mathbf{u})$ be some function supported at $[-1/2, 1/2]^d$, $\phi(\mathbf{u}) \geq 0$, and for any $\mathbf{u}, \mathbf{u}'$,

$$|\phi(\mathbf{u}) - \phi(\mathbf{u}')| \leq C_b \|\mathbf{u} - \mathbf{u}'\|_2^\beta. \tag{221}$$

Suppose $\mathbf{c}_1, \ldots, \mathbf{c}_B$ be the centers of $B$ bins, $f(\mathbf{x}) = 1$, and

$$\eta(\mathbf{x}) = \sum_{k=1}^{B} v_k \phi\left(\frac{\mathbf{x} - \mathbf{c}_k}{l}\right) l^\beta, \tag{222}$$

in which $v_k \in \{-1, 1\}$. Then let

$$\hat{v}_k = \arg\min_{s \in \{-1,1\}} \int_{B_k} \left(\hat{\eta}(\mathbf{x}) - s\phi\left(\frac{\mathbf{x} - \mathbf{c}_k}{l}\right) l^\beta\right)^2 f(\mathbf{x}) d\mathbf{x}. \tag{223}$$

Then

$$\begin{aligned}
R &= \mathbb{E}\left[\int (\hat{\eta}(\mathbf{x}) - \eta(\mathbf{x}))^2 f(\mathbf{x}) d\mathbf{x}\right] \\
&= \sum_{k=1}^{B} \mathbb{E}\left[\int_{B_k} (\hat{\eta}(\mathbf{x}) - \eta(\mathbf{x}))^2 f(\mathbf{x}) d\mathbf{x}\right] \\
&\overset{(a)}{\geq} \sum_{k=1}^{B} l^{2\beta+d} \|\phi\|_2^2 \, \mathrm{P}(\hat{v}_k \neq v_k) \\
&= \|\phi\|_2^2 \, l^{2\beta+d} \mathbb{E}[\rho_H(\hat{\mathbf{v}}, \mathbf{v})].
\end{aligned} \tag{224}$$

Here we explain (a). Without loss of generality, suppose $v_k = -1$, $\hat{v}_k = 1$. Then

$$\int_{B_k} \left(\hat{\eta}(\mathbf{x}) - \phi\left(\frac{\mathbf{x} - \mathbf{c}_k}{l}\right) l^\beta\right)^2 f(\mathbf{x}) d\mathbf{x} \leq \int_{B_k} \left(\hat{\eta}(\mathbf{x}) + \phi\left(\frac{\mathbf{x} - \mathbf{c}_k}{l}\right) l^\beta\right)^2 f(\mathbf{x}) d\mathbf{x}. \tag{225}$$

Note that

$$\begin{aligned}
&\int_{B_k} \left(\hat{\eta}(\mathbf{x}) - \phi\left(\frac{\mathbf{x} - \mathbf{c}_k}{l}\right) l^\beta\right)^2 f(\mathbf{x}) d\mathbf{x} + \int_{B_k} \left(\hat{\eta}(\mathbf{x}) + \phi\left(\frac{\mathbf{x} - \mathbf{c}_k}{l}\right) l^\beta\right)^2 f(\mathbf{x}) d\mathbf{x} \\
&= 2\int_{B_k} \left(\hat{\eta}^2(\mathbf{x}) + \phi^2\left(\frac{\mathbf{x} - \mathbf{c}_k}{l}\right) l^{2\beta}\right) f(\mathbf{x}) d\mathbf{x} \\
&\geq 2l^{2\beta+d} \|\phi\|_2^2.
\end{aligned} \tag{226}$$

Thus

$$\int_{B_k} \left( \hat{\eta}(\mathbf{x}) + \phi\left(\frac{\mathbf{x} - \mathbf{c}_k}{l}\right) l^\beta \right)^2 f(\mathbf{x})d\mathbf{x} \geq l^{2\beta+d} \|\phi\|_2^2. \tag{227}$$

Similar bound holds if $v_k = 1$ and $\hat{v}_k = -1$. Now (a) in (224) has been proved. From (224),

$$\inf_{\hat{\eta}} \sup_{(f,\eta)\in\mathcal{P}_{reg}} R \geq \|\phi\|_2^2 l^{2\beta+d} \inf_{\hat{\mathbf{v}}} \sup_{\mathbf{v}} \mathbb{E}[\rho_H(\hat{\mathbf{v}}, \mathbf{v})]. \tag{228}$$

Define

$$\delta = \max_{\mathbf{v},\mathbf{v}':\rho_H(\mathbf{v},\mathbf{v}')=1} D(p_{\mathbf{Z}|\mathbf{v}}||p_{\mathbf{Z}|\mathbf{v}'}). \tag{229}$$

Follow the analysis of nonparametric classification, let

$$l \sim (nm\epsilon^2)^{-\frac{1}{2(d+\beta)}}, \tag{230}$$

then $\delta \lesssim 1$. Hence, By (Tsybakov, 2009), Theorem 2.12(iv),

$$\inf_{\hat{\mathbf{v}}} \sup_{\mathbf{v}} \mathbb{E}[\rho_H(\hat{\mathbf{v}}, \mathbf{v})] \gtrsim B \sim l^{-d}, \tag{231}$$

hence from (228),

$$\inf_{\hat{\eta}} \sup_{(f,\eta)\in\mathcal{P}_{reg}} R \gtrsim l^{2\beta+d} \cdot l^{-d} = h^{2\beta} \sim (nm\epsilon^2)^{-\frac{\beta}{d+\beta}}. \tag{232}$$

# G AUXILIARY LEMMAS

**Lemma 10.** *Suppose there are two probability measures $p_1$ and $p_2$ supported at $\mathcal{X}$. $p_1 = p_2$ except at $S \subset \mathcal{X}$. Then*

$$\mathbb{TV}(p_1, p_2) \leq \sqrt{\frac{1}{2} p_1(S) D(p_1||p_2)}. \tag{233}$$

*Proof.* Denote $\mathbb{E}_1$ as the expectation under $p_1$. Denote $p_{1|S}$ and $p_{2|S}$ as the conditional distribution of $p_1$ and $p_2$ on $S$.

$$
\begin{aligned}
D(p_1||p_2) &= \mathbb{E}_1\left[\ln\frac{p_1}{p_2}\right] \\
&= p_1(S)\mathbb{E}_{1|S}\left[\ln\frac{p_{1|S}}{p_{2|S}}\right] \\
&\geq 2p_1(S)\mathbb{TV}^2(p_{1|S}, p_{2|S}) \\
&= 2p_1(S)\left[\frac{\mathbb{TV}(p_1, p_2)}{p_1(S)}\right]^2 \\
&= \frac{2\mathbb{TV}^2(p_1, p_2)}{p_1(S)}. 
\end{aligned} \tag{234}
$$

The proof is complete. $\square$

**Lemma 11.** *Under Assumption 2(a), there exists a constant $C$, such that for any $s > 0$,*

$$\mathbb{E}\left[|\eta(\mathbf{X})|e^{-s|\eta(\mathbf{X})|^2}\right] \leq Cs^{-\frac{1}{2}(\gamma+1)}. \tag{235}$$

*Proof.*

$$
\begin{aligned}
\mathbb{E}\left[|\eta(\mathbf{X})|e^{-s|\eta(\mathbf{X})|^2}\right] &= \mathbb{E}\left[|\eta(\mathbf{X})|e^{-\frac{s}{2}|\eta(\mathbf{X})|^2}e^{-\frac{s}{2}|\eta(\mathbf{X})|^2}\right] \\
&\leq \left(\sup_{u\geq 0}ue^{-\frac{s}{2}u^2}\right)\mathbb{E}\left[e^{-\frac{s}{2}|\eta(\mathbf{X})|^2}\right] \\
&= \frac{1}{\sqrt{s}}e^{-\frac{1}{2}}\mathbb{E}\left[e^{-\frac{s}{2}|\eta(\mathbf{X})|^2}\right] \\
&= \frac{1}{\sqrt{s}}e^{-\frac{1}{2}}\int_0^1 \mathrm{P}\left(e^{-\frac{s}{2}|\eta(\mathbf{X})|^2} > t\right)dt \\
&= \frac{1}{\sqrt{s}}e^{-\frac{1}{2}}\int_0^1 \mathrm{P}\left(|\eta(\mathbf{X})| < \sqrt{\frac{2\ln\frac{1}{t}}{s}}\right)dt \\
&\leq \frac{C_a}{\sqrt{s}}e^{-\frac{1}{2}}\int_0^1 \left(\frac{2\ln\frac{1}{t}}{s}\right)^{\frac{\gamma}{2}}dt \\
&\leq 2^{\frac{\gamma}{2}}C_a e^{-\frac{1}{2}}s^{-\frac{1+\gamma}{2}}\Gamma\left(\frac{\gamma}{2}+1\right),
\end{aligned}
\tag{236}
$$

in which $\Gamma(u) = \int_0^\infty t^{u-1}e^{-t}dt$ is the Gamma function. $\qquad\square$

