# OpenReview forum: "Learning with User-Level Local Differential Privacy"
_ICLR.cc/2025/Conference — Submitted to ICLR 2025_

### Official Review · Reviewer_mdCH · 2024-11-03

**Soundness:** 2
**Presentation:** 3
**Contribution:** 2
**Rating:** 3
**Confidence:** 4

**Summary:**

This paper addresses the problem of achieving user-level local differential privacy (LDP) across various statistical tasks, including mean estimation, stochastic optimization, classification, and regression. By tailoring privacy mechanisms to different privacy levels, the authors propose algorithms that attain optimal performance rates under user-level LDP, achieving minimax optimality up to logarithmic factors. Unlike the central model, where user-level privacy often implies slower convergence, the local model yields convergence rates comparable to item-level LDP, with even faster rates in heavy-tailed distributions. This work provides both theoretical bounds and adaptive strategies, expanding the scope of user-level LDP applications in distributed systems.

**Strengths:**

+ It addressed user-level DP, which is a relatively less explored but extremely relevant area
+ It studied a wide variety of tasks (mean estimation, stochastic optimization, nonparametric classification and regression)

**Weaknesses:**

- Technical novelty is unclear
- Some proof is unclear

**Questions:**

1. The authors highlight the regime where \eps > 1 in the introduction. Yet, it is unclear how this regime is handled in the proof of the lower bound. For example, in the proof of Theorem 2, how do we get from Eq. 62 to Eq. 64, if \eps > 1? I understand that the current proof holds for \eps < 1. Similar questions exist in the proof of Theorems 6 and 7.

2. Following the above, it would be useful to highlight the difference in the lower bound proof for item-level and user-level LDP, especially the regime when \eps >1.

3. It seems to me that the current local model is **non-interactive**? Can the authors comment on the **interactive** model? That is, can the proposed algorithms be easily extended to the interactive model?

---

> ### Author Response · Authors · 2024-11-19
> **Response**
>
> Thanks for your comment. Now we respond to your questions.
>
> # Reply to Question 1
>
> If $\epsilon > 1$, then we just use eq.(65), which is the standard minimax lower bound that holds even for non-private data. We do not need to derive eq.(64) from eq.(62) again. eq.(64) only holds for $\epsilon < 1$. Therefore our proof is correct here.
>
> Following your questions, in our revised paper, we have emphasized this point to make it clearer. The new statements are:
>
> With $\epsilon<1$, let $s\sim 1/\sqrt{nm\epsilon^2}$, then $\text{P}(\hat{V}\neq V)\sim 1$. Hence
>
> ```
> \inf_{\hat{\mu}} \underset{Q\in \mathcal{Q}_\epsilon}{\inf} \underset{p\in \mathcal{P}_\mathcal{X}}{\sup}\mathbb{E}[(\hat{\mu}-\mu)^2]\gtrsim \frac{D^2}{nm\epsilon^2}
> ```
>
> If $\epsilon>1$, then from standard minimax analysis for non-private problems, \textcolor{blue}{the estimation error can not be smaller than $\sigma^2/(mn)$, with $\sigma^2$ being the sample variance. The maximum value of $\sigma^2$ is $D^2$. Therefore} it can be easily shown that
>
> ```
> \underset{\hat{\mu}}{\inf} \underset{Q\in \mathcal{Q}_\epsilon}{\inf}\underset{p\in \mathcal{P}_\mathcal{X}}{\sup}\mathbb{E}[(\hat{\mu}-\mu)^2]\gtrsim \frac{D^2}{nm}.
> ```
>
> # Reply to Question 2
>
> Thanks for this suggestion. To derive the lower bound, we need to bound the total variation (TV) distance, which is shown in Lemma 10 in Appendix G.
>
> # Reply to Question 3
>
> Current model is actually **interactive**. For example, in the mean estimation problem, we use a two-stage approach, in which the second stage depends on the first stage.
>
> # About the novelty of this paper
>
> To the best of our knowledge, our work is the first attempt to study user-level LDP problems for **general $\epsilon$**. Moreover, it is also the first attempt to study nonparametric classification and regression problem with user-level LDP.
>
> For user-level LDP problems, different privacy budget $\epsilon$ requires different methods. We take the multi-dimensional mean estimation problem as an example. We conduct **user splitting** (divide users into groups, and each group is responsible for one component) for small $\epsilon$, and **budget splitting** (let the budget to be $\epsilon / d$ for each component) for very large $\epsilon$. For medium $\epsilon$, we design a new grouping strategy to achieve a smooth transition between these two extremes.
>
> We are looking forward to your further comments and questions.

---

### Official Review · Reviewer_Qvtb · 2024-11-04

**Soundness:** 3
**Presentation:** 3
**Contribution:** 3
**Rating:** 6
**Confidence:** 4

**Summary:**

This paper examines user-level privacy in a distributed setting, particularly in user-level local differential privacy (ULDP). The authors analyze mean estimation and its applications in stochastic optimization, classification, and regression, proposing adaptive strategies that optimize performance across various privacy levels. The authors claim that unlike in the central model, the convergence rates for user-level and item-level privacy are nearly equivalent in local models, with user-level privacy yielding even faster rates for heavy-tailed distributions.

**Strengths:**

1. The paper is very organized and presents its results in a clear manner.

2. Matching information-theoretic lower bounds are also derived which enhances the completeness of this work.

**Weaknesses:**

This paper studies ULDP on various problem settings: mean estimation, stochastic optimization, classification and regression. It is clear from Table 1 how the proposed rates in ULDP is different from the rates in item-level LDP.  However, some relevant papers appear to be missing from the references. For example, [1], [2] and [3]

[1]: Li, Bo, Wei Wang, and Peng Ye. "Improved Bounds for Pure Private Agnostic Learning: Item-Level and User-Level Privacy." arXiv preprint arXiv:2407.20640 (2024).

[2]: Cummings, Rachel, et al. "Mean estimation with user-level privacy under data heterogeneity." Advances in Neural Information Processing Systems 35 (2022): 29139-29151.

[3]: Charles, Zachary, et al. "Fine-tuning large language models with user-level differential privacy." arXiv preprint arXiv:2407.07737 (2024).


Besides, on line 132 and 133" Moreover, we also provide the first analysis on nonparametric classification and regression problems under user-level ϵ-LDP" is not accurate. To the best of my knowledge, [4] also studies regression in the ULDP setting under sparsity constraint. From my perspective, sparse estimation problem in LDP model ([5], [6]) might also could also be a valuable addition to the related work section.

[4]: Ma, Yuheng, Ke Jia, and Hanfang Yang. "Better Locally Private Sparse Estimation Given Multiple Samples Per User." arXiv preprint arXiv:2408.04313 (2024).

[5]: Zhu, Liyang, et al. "Improved Analysis of Sparse Linear Regression in Local Differential Privacy Model." arXiv preprint arXiv:2310.07367 (2023).

[6]: Zhou, Mingxun, et al. "Locally differentially private sparse vector aggregation." 2022 IEEE Symposium on Security and Privacy (SP). IEEE, 2022.

**Questions:**

Please see above.

---

> ### Author Response · Authors · 2024-11-19
> **Response**
>
> Thank you very much for your positive feedback. We are encouraged that you have positive evaluation on our significance, organization and completeness of our work.
>
> The references [1-3] are about user-level DP under the **central model**. Although these works and ours do not have exactly the same background, we will definitely mention them in our revised paper.
>
> To the best of our knowledge, the statement "Moreover, we also provide the first analysis on nonparametric classification and regression problems under user-level $\epsilon$-LDP" is still accurate. [4] studies linear regression instead of nonparametric classification and regression. Since nonparametric statistics do not impose any model assumptions on the distribution, the techniques are crucially different. [5] and [6] are about item-level LDP. However, following your comment, we will discuss them in the related work section.

---

> > ### Comment · Reviewer_Qvtb · 2024-11-26
> >
> > Thank you for the response and I have no other comments.

---

### Official Review · Reviewer_skTW · 2024-11-04

**Soundness:** 2
**Presentation:** 2
**Contribution:** 2
**Rating:** 5
**Confidence:** 3

**Summary:**

This paper first analyzes the mean estimation problem and then extends the findings to stochastic optimization, classification, and regression. Specifically, the authors propose adaptive strategies to achieve optimal performance across all privacy levels. They also derive information-theoretic lower bounds, demonstrating that the proposed methods are minimax optimal up to logarithmic factors. Notably, unlike the central DP model, where user-level DP generally leads to slower convergence, the results show that, under the local DP model, convergence rates are nearly identical between user-level and item-level cases for distributions with bounded support. For heavy-tailed distributions, the user-level rate is even faster than the item-level rate.

**Strengths:**

The paper tackles significant learning problems under user-level local differential privacy (LDP) constraints and establishes several tight lower and upper bounds.

**Weaknesses:**

Some statements throughout the paper are somewhat unclear, which can make parts of the presentation difficult to follow.

For the stochastic optimization problem, only the bounded gradient case and strongly convex objective functions are considered, which may not be sufficiently practical for broader applications.

**Questions:**

1. Why is the case of $\epsilon > 1$ considered interesting for LDP studies?

2. In Proposition 1 (2), to ensure user-level $\epsilon$-LDP from item-level $\epsilon$-LDP, if we randomly pick a sample from each user, why is it stated as ''$n$ users with $m$ samples per user'' instead of ''$n$ users with $1$ sample per user''?

3. For Definition 1, could you explain in detail why the definition of user-level $\epsilon$-LDP does not ensure item-level $\epsilon$-LDP?

4. For Theorem~1, I am unable to understand why is it said that the mean squared error will never converge to zero with increasing $m$ if $n$ is fixed.

5. What does $n_0$ represent in Equation (6)?

6. For the stochastic optimization problem, why is only the bounded gradient case considered? Why can't the private mean estimation over unbounded support developed in the paper be used for the unbounded gradient case, which seems more interesting and important in practice?

---

> ### Author Response · Authors · 2024-11-19
> **Response**
>
> Thanks for your careful reading of this paper and valuable comments. We reply to your comments as follows.
>
> # Reply to Question 1
>
> In industrial applications, from an accuracy-first perspective, companies usually consider only the case with $\epsilon >1$. Moreover, a recently trend is to use message shuffling techniques. In these cases, we usually consider relatively large $\epsilon$. For example:
>
> K.Talwar et al. Samplable Anonymous Aggregation for Private Federated Data Analysis, CCS, 2024. (apple team)
>
> Exposure Notification Privacy-Preserving. "Exposure Notification Privacy-Preserving Analytics (ENPA) White Paper." ENPA_White_Paper. pdf.
>
> # Reply to Question 2
>
> We mean that, for arbitrary $m$, if we randomly pick a sample from a user, $n$ users with $m$ samples per user under user-level $\epsilon$-LDP is just equivalent to item-level $\epsilon$-LDP with $n$ samples. The result is not limited to the case with $m=1$.
>
> # Reply to Question 3
>
> For user-level $\epsilon$-LDP, $m$ local samples combine together to generate an output, thus **local samples within the same user can share information with each other.** Therefore user-level $\epsilon$-LDP does not ensure item-level $\epsilon$-LDP.
>
> # Reply to Question 4
>
> We explain it from information-theoretic perspective. A precise communication of a continuous random variable requires infinite number of bits. However, in user-level LDP, each user compresses all local samples into only one variable, which can only transmit limited information. As a result, as long as $n$ is fixed, no matter how large is $m$, the precision can not reach infinity. We refer to line 251-252 in our paper, which gives a lower bound. We also refer to line 969-986 for a proof.
>
> # Reply to Question 5
>
> In line 322, our paper has stated that "denote $n_0$ as the number of users in each group". Now there are $n$ users divided into $\lceil d/\epsilon \rceil$ groups, thus $n_0=n/\lceil d/\epsilon\rceil$.
>
> # Reply to Question 6
>
> Thanks for this comment. We study the bounded gradient case because we want to compare with the item-level case (Duchi et al. .Local privacy and statistical minimax rates. FOCS 2013). We will add the analysis for unbounded gradients in our revised paper.

---

> > ### Comment · Reviewer_skTW · 2024-11-27
> > **Response**
> >
> > Thank you for your detailed responses to my questions. While the clarifications provided are helpful and address several points, some concerns regarding the practicality of the bounded gradient assumption and the presentation's clarity remain.
> >
> > I maintain my score, as the paper is marginal in its present form. I look forward to seeing the revised manuscript and the planned extensions.

---

### Author Response · Authors · 2024-11-22
**Global Response**

We thank the reviewers for your time in reviewing this paper, as well as your valuable comments. We are encouraged that reviewers agree that our paper is complete and well organized (Reviewer Qvtb) tackle a significant problem (Reviewer sKTW). There are also some detailed comments that help us to improve the paper. Following these comments, we have revised the paper and the revisions are marked with **blue** color.

We respond to your comments below. Hope that these responses together with the revised manuscript can address your concerns. We are also looking forward to the reevaluation of our paper.

---

### Meta-Review · Area_Chair_jD1Z · 2024-12-14

**Metareview:**

## Summary of Contributions

This paper studies learning tasks in user-level local DP (LDP) setting. Here each of the $n$ users has $m$ items drawn from an unknown distribution, and we want to satisfy LDP where the privacy unit is each user (i.e. all $m$ items can change). The authors study this model for the problem of mean estimation, stochastic optimization, classification and regression. They prove nearly tight (minimax) bounds for these problem for all ranges of $\epsilon$ values.

## Strengths

- The problems and the user-level DP settings are natural and important.
- The bounds obtained here are nearly tight.

## Weaknesses

- It is unclear how novel this work is. The (Bassily & Sun, 2023) paper cited here already obtain tight bounds for small $\epsilon$ values, which is arguably the most important regime of parameters. Similarly, the techniques in this paper are fairly similar to that paper (i.e. dividing user into groups, rotation for $\ell_2$ vectors) and it seems like the only main differences are some sort of parameter tuning / rearranging of different steps. (Note that the novelties are not clearly highlighted in the paper.)

## Recommendation

Given the weakness, we believe that the paper's contributions are below the bar for ICLR and we recommend rejection.

**Additional Comments On Reviewer Discussion:**

There are some clarification with regards to novelty and relation to previous work during discussion. However, it remains true that the algorithms are very minor tweak of previous algorithms and thus the novelty remains unclear.

---

### Decision · Program_Chairs · 2025-01-22

Reject